

# Landau levels, response functions and magnetic oscillations from a generalized Onsager relation

**Jean-Noël Fuchs[1,2]\*, Frédéric Piéchon[2] and Gilles Montambaux[2]**

**1** Sorbonne Université, CNRS, Laboratoire de Physique Théorique de la Matière Condensée, F-75005 Paris, France
**2** Laboratoire de Physique des Solides, CNRS, Université Paris-Sud, Université Paris-Saclay, F-91405 Orsay, France

\* fuchs@lptmc.jussieu.fr

## Abstract

A generalized semiclassical quantization condition for cyclotron orbits was recently proposed by Gao and Niu [1], that goes beyond the Onsager relation [2]. In addition to the integrated density of states, it formally involves magnetic response functions of all orders in the magnetic field. In particular, up to second order, it requires the knowledge of the spontaneous magnetization and the magnetic susceptibility, as was early anticipated by Roth [3]. We study three applications of this relation focusing on two-dimensional electrons. First, we obtain magnetic response functions from Landau levels. Second we obtain Landau levels from response functions. Third we study magnetic oscillations in metals and propose a proper way to analyze Landau plots (i.e. the oscillation index $n$ as a function of the inverse magnetic field $1/B$) in order to extract quantities such as a zero-field phase-shift. Whereas the frequency of $1/B$-oscillations depends on the zero-field energy spectrum, the zero-field phase-shift depends on the geometry of the cell-periodic Bloch states via two contributions: the Berry phase and the average orbital magnetic moment on the Fermi surface. We also quantify deviations from linearity in Landau plots (i.e. aperiodic magnetic oscillations), as recently measured in surface states of three-dimensional topological insulators and emphasized by Wright and McKenzie [4].



# 1 Introduction

The quantization of closed cyclotron orbits for Bloch electrons in the presence of a magnetic field leads to the formation of Landau levels (LLs) [5]. When the Fermi level crosses the LLs, physical quantities such as the resistance, the magnetization or the density of states feature quantum magnetic oscillations [6]. The analysis of the latter is considered as a powerful way of extracting informations on the band structure of metals [7]. Band structure here refers to both the zero-field band energy spectrum $\epsilon_n(\mathbf{k})$ and to the geometry of the cell-periodic Bloch states $|u_n(\mathbf{k})\rangle$. The standard way to analyze magnetic oscillations is to use Onsager's quantization condition [2] to obtain Landau levels for Bloch electrons and the Lifshitz-Kosevich formula [8] that describes the temperature and disorder dependence of the amplitude of oscillations.

As early as 1966, Roth generalized Onsager's condition by including inter-band effects in the semiclassical quantization condition of closed cyclotron orbits up to second order in the magnetic field [3] (see also Appendix B for other important contributions). In a recent insightful paper, Gao and Niu [1] proposed a further extension by systematically including higher-order corrections in the magnetic field in a compact and thermodynamic manner. Their equation generalizes Onsager's relation [2]

$$(n+\frac{1}{2})\frac{eB}{h} = N_0(\epsilon), \tag{1}$$

which relates $N_0(\epsilon)$, the zero-field integrated density of states (IDoS) at the Fermi energy $\epsilon$, to the degeneracy $eB/h$ of a LL, where $B > 0$ is the modulus of the magnetic field, and the Landau index $n$, which is an integer [here we assumed a sample area $\mathscr{A} = 1$]. The relation

obtained by Gao and Niu reads

$$(n + \frac{1}{2})\frac{eB}{h} = N(\epsilon, B) = N_0(\epsilon) + BM_0'(\epsilon) + \frac{B^2}{2}\chi_0'(\epsilon) + \sum_{p \geq 3} \frac{B^p}{p!} R_p'(\epsilon), \tag{2}$$

where $N(\epsilon, B)$ is the smoothed (i.e. without the magnetic oscillations[1]) IDoS in the presence of a magnetic field $B$, $M_0(\epsilon)$ is the spontaneous magnetization, $\chi_0(\epsilon)$ is the magnetic susceptibility, $R_p'$ are higher order magnetic response functions and the prime denotes the derivative with respect to the energy $M_0' = \partial_\epsilon M_0$. These quantities $N_0$, $M_0'$, $\chi_0'$, $R_p'$ are taken at zero temperature and in the limit of zero magnetic field, and depend on the Fermi energy $\epsilon$. The conceptual important difference between (1) and (2) is that whereas the first relies only on the zero field energy spectrum (or more precisely on the zero-field IDoS), the supplementary terms in the second explicitly involve information on the zero field cell-periodic Bloch states and on interband coupling. This extra information is carried by quantities such as the magnetization and the susceptibility. Indeed, the magnetization involves the orbital magnetic moment and the Berry curvature [9], and the susceptibility involves not only the curvature of the energy spectrum but also the Berry curvature and the quantum metric [10].

Interestingly equation (2) may be rewritten in a form similar to (1)

$$[n + \gamma(\epsilon, B)]\frac{eB}{h} = N_0(\epsilon), \tag{3}$$

but involving an energy and magnetic-field dependent phase-shift

$$\gamma(\epsilon, B) = \frac{1}{2} - \frac{h}{e}M_0'(\epsilon) - \frac{hB}{2e}\chi_0'(\epsilon) - \sum_{p \geq 3} \frac{hB^p}{ep!} R_p'(\epsilon), \tag{4}$$

instead of a mere constant $\frac{1}{2}$. Roth wrote a formally similar equation (see (42) in [3]), however she did not relate the first (resp. second) order correction to the derivative of the magnetization (resp. magnetic susceptibility) and did not obtain the higher-order corrections in the field. In the following, we will call eq. (2) the Roth-Gao-Niu relation.

There are at least three important consequences of this generalized Onsager quantization condition:

1) When LLs are known analytically, their expression can be inverted to obtain response functions analytically (actually their derivatives $\frac{\partial M_0}{\partial \epsilon}$ and $\frac{\partial \chi_0}{\partial \epsilon}$).

2) Response functions can be used to obtain LLs in cases where the latter are hard to obtain exactly and for which response functions are known by other means (i.e. linear response theory).

3) The phase of magnetic oscillations (related to $\gamma(\epsilon, B)$), such as Shubnikov-de Haas oscillations in the longitudinal resistance or de Haas-van Alphen oscillations in the magnetization, can be derived from the Roth-Gao-Niu relation. This helps to analyze Landau plots (i.e. index $n$ of oscillations as a function of the inverse magnetic field $1/B$). In particular, we can obtain a simple formula for the zero-field phase-shift $\gamma(\epsilon, 0)$ in terms of the Maslov index, the Berry phase and the orbital magnetic moment (averaged over the Fermi surface). Also we can see the general structure of the $\gamma(\epsilon, B)$ and study deviations from linear Landau plots, i.e. aperiodic magnetic oscillations, as recently measured in topological insulator surface states and discussed by Wright and McKenzie [4].

The aim of the present paper is to elaborate on these three consequences. The article is organized as follows. We first review the Roth-Gao-Niu quantization condition and its validity (section 2), then present the three type of consequences: from LLs to response functions

---

[1]The smoothed IDoS is called semiclassical IDoS by Gao and Niu [1]. By definition it is the IDoS without the magnetic oscillations. See the discussion in Appendix A.

(section 3) on the examples of the graphene monolayer and bilayer, a generic 2D semiconductor and the Rashba model; from response functions to LLs (section 4) on the examples of the Hofstadter, the semi-Dirac and the bilayer models; and an analysis of magnetic oscillations in section 5. Eventually, we give a conclusion in section 6. Some material is also presented in appendices.

## 2  Roth-Gao-Niu quantization condition

We start by shortly reviewing the Roth-Gao-Niu quantization condition and then discuss its validity. It is argued [1] that for a half-filled LL (i.e. when the Fermi energy $\epsilon$ is exactly at the energy $\epsilon_n$ of the $n$th LL) the number of states below the Fermi level at zero temperature is

$$(n + \frac{1}{2})B = N(\epsilon, B) = -\Omega'(\epsilon, B, T = 0), \tag{5}$$

where $\Omega(\epsilon, B, T)$ is the non-oscillatory part (the smooth part) of the grand potential and the prime denotes the partial derivative with respect to the chemical potential $\Omega' = \partial_\epsilon \Omega$. Here, we assumed that the sample area $\mathscr{A} = 1$ so that the degeneracy of a LL is $N_\phi = \frac{eB.\mathscr{A}}{h} = B$ with $\hbar = 1$ and $e = 2\pi$ such that the flux quantum $\phi_0 = h/e = 1$. The temperature is assumed to be larger than the typical splitting between LLs: $T \gg \omega_c \propto B$ with $T \to 0$ and $B \to 0$. This requirement ensures that one is working with the low field expansion of the smooth (non-oscillatory) part of the grand potential (see the discussion in Appendix A). Note that the above relation is far from obvious as the left hand side denotes the quantum mechanical IDoS at zero temperature, while the right hand side is for the smoothed IDoS, typically obtained at temperature higher than the separation between LLs.

It is assumed that the smooth grand potential $\Omega$ may be written as a power series in $B$ (see below for a discussion of the validity of this assumption). Its expansion is written as

$$\Omega(\epsilon, B) = \Omega_0(\epsilon) - BM_0(\epsilon) - \frac{B^2}{2}\chi_0(\epsilon) - \sum_{p \geq 3} \frac{B^p}{p!} R_p(\epsilon), \tag{6}$$

where $M_0$ is the zero-field (spontaneous) magnetization, $\chi_0$ is the zero-field magnetic susceptibility and $R_p(\epsilon) = -\frac{\partial^p \Omega}{\partial B^p}(\epsilon, B = 0)$ with $p \geq 3$ are higher order magnetic response functions (all at zero temperature). From the expansion, it follows that

$$(n + \frac{1}{2})B = N_0(\epsilon) + BM_0'(\epsilon) + \frac{B^2}{2}\chi_0'(\epsilon) + \sum_{p \geq 3} \frac{B^p}{p!} R_p'(\epsilon), \tag{7}$$

where $N_0(\epsilon) = -\Omega_0'(\epsilon)$ is the zero-field IDoS. This is the Roth-Gao-Niu relation [1]. When keeping only $N_0(\epsilon)$ in the r.h.s., it reduces to the Onsager quantization condition [2]. When keeping first order correction to the Onsager relation, $N_0(\epsilon) = (n + \frac{1}{2} - M_0'(\epsilon))B$, it shows the appearance of Berry phase type of corrections – hidden in $M_0'$ – and recovers various results scattered in the literature. When keeping second order corrections, $N_0(\epsilon) = (n + \frac{1}{2} - M_0'(\epsilon))B - \chi_0'(\epsilon)\frac{B^2}{2}$, it is formally equivalent to the Roth quantization condition [3], albeit written in a much more compact and transparent form. This will be discussed in detail in section V.

We now discuss the validity of the Roth-Gao-Niu relation. There are several issues:

(i) A first issue concerns its application to a system with degeneracies such as several valleys or spin projections. In fact, it is only meaningful for each species separately [1]. This implies that on the right hand side of Eq.(7), the effective zero field quantities $N_0(\epsilon), M_0'(\epsilon)$ and $\chi_0'(\epsilon)$ are species dependent and thus do not correspond to directly measurable equilibrium

thermodynamic responses. In particular, for time reversal invariant systems, there is no finite thermodynamic spontaneous magnetization but the species dependent effective spontaneous magnetization contribution $M'_0(\epsilon)$ might nevertheless appear finite.

(ii) Equation (7) is valid for electron-like but not for hole-like cyclotron orbits. In the latter case, it becomes

$$(n+\frac{1}{2})B = N_{\text{tot}} - N(\epsilon, B) = N_{\text{tot}} - N_0(\epsilon) - BM'_0(\epsilon) - \frac{B^2}{2}\chi'_0(\epsilon) - \sum_{p\geq 3}\frac{B^p}{p!}R'_p(\epsilon), \qquad (8)$$

where $N_{\text{tot}}$ is the total number of electrons when the band is full and $n \in \mathbb{N}$. Actually $N(\mu, B) = \int d\epsilon \rho(\epsilon, B) f(\epsilon, \mu)$ in (7) is replaced by $N_{\text{tot}} - N(\mu, B) = \int d\epsilon \rho(\epsilon, B)[1 - f(\epsilon, \mu)]$ in (8), i.e. upon substituting the Fermi function $f$ by $1 - f$, where $\rho(\epsilon, B)$ is the DoS. Generally speaking $N_{\text{tot}}$ is a constant that is either determined by the occupation of a full band in the case of a model defined on a lattice or should be determined from self-consistency in the case of an effective low-energy model. See also Appendix D where the Onsager quantization conditions for hole orbits is discussed.

(iii) Semiclassical quantization condition for closed cyclotron orbits (either Roth-Gao-Niu or Onsager) predicts perfectly degenerate Landau levels and does not account for lattice broadening of Landau levels into bands [11]. Indeed, it does not include tunneling between cyclotron orbits belonging either to different valleys or to another Brillouin zone, i.e. magnetic breakdown [12]. All these phenomena are beyond the present description. Neglecting magnetic breakdown is consistent with the assumption that the grand potential admits a series expansion in (positive) integer powers of $B$ and does not contain terms such as $e^{-\#/B}$.

(iv) The Roth-Gao-Niu relation fails to capture singular behaviors in response functions, that may appear at specific energies corresponding to band contacts or edges. More precisely, it misses step functions or delta functions in $N_0$, $M_0$, $\chi_0$, etc. As a first example, the Roth-Gao-Niu relation does not account for the McClure diamagnetic delta-peak in the susceptibility at the Dirac point of graphene [13]. Similarly for massive Dirac fermions (gapped graphene or boron nitride), the step functions at the gap edges of the diamagnetic susceptibility plateau [14] are not accounted for by the present formalism. In the first example, at the energy of the band contact, the grand potential actually behaves as $B^{3/2}$ and therefore does not admit an expansion in integer powers of $B$, as is assumed in the Roth-Gao-Niu relation.

(v) The right-hand side of eq. (2) is an asymptotic series in powers of $B$. Here we stress that the small parameter of this expansion is $B$ at fixed $(n+\frac{1}{2})B$. The fact that the left-hand side of eq. (2) is fixed comes from the quantization occurring at constant energy $\epsilon$ and from Onsager's relation $(n+\frac{1}{2})B \approx N_0(\epsilon) = $ constant. Because $(n+\frac{1}{2})B$ is fixed, the small parameter $B$ can also be thought as $\frac{1}{n+1/2}$, where one recognizes the usual semi-classical criterion of large $n$.

## 3 From Landau levels to magnetic response functions

When available, one may use the knowledge of the exact LLs to obtain magnetic response functions such as the magnetization and the susceptibility. In some cases, it may be easier to proceed that way rather than to try to compute directly these response function using linear response theory. In the following, we treat four examples in detail: (1) gapped graphene monolayer, (2) gapped graphene bilayer, (3) gapped Dirac electrons for a semiconductor and (4) the Rashba model. The three first examples are gapped and have a valence and a conduction band. The fourth example has a Fermi surface that can be electron or hole-like. It is therefore essential to be able to treat both electrons and holes so that the Roth-Gao-Niu relation should be adapted to account both for electrons and holes contributions. It is thus

convenient to regroup equations (7) and (8) into a single relation

$$n = \frac{sN_0(\epsilon) + N_{\text{tot}}\Theta(-s)}{B} - \frac{1}{2} + sM'_0(\epsilon) + \frac{B}{2}s\chi'_0(\epsilon) + ..., \tag{9}$$

where $s = \text{sign}(\epsilon)$ is the sign of the energy and $sN_0(\epsilon) + N_{\text{tot}}\Theta(-s)$ is simply the IDoS of either electrons $N_{\text{el}}(\epsilon) = N_0(\epsilon)$ or holes $N_{\text{h}}(\epsilon) = N_{\text{tot}} - N_0(\epsilon)$ depending on $s$. It gives the Landau index $n$ as a Laurent series in the magnetic field $B$ starting with a $1/B$ term: $n = \sum_{p=-1}^{\infty} c_p B^p$.

The main result of this section is that the Roth-Gao-Niu relation indeed allows one to recover many results concerning the energy-derivative of magnetic response functions. However, it fails to recover singular behaviors such as step functions, Dirac delta functions, etc.

### 3.1 Gapped graphene monolayer with Zeeman effect

We consider the low energy description of a gapped graphene monolayer (e.g. boron nitride) in the presence of a Zeeman effect. The Hamiltonian (with $\xi = \pm 1$ the valley index) is [20]

$$H_\xi = (\nu\xi\Pi_x\tau_x + \nu\Pi_y\tau_y + \Delta\tau_z)\sigma_0 + \Delta_Z\sigma_z\tau_0, \tag{10}$$

where $\boldsymbol{\Pi} = \boldsymbol{p} + 2\pi\boldsymbol{A}$ is the gauge-invariant momentum, $\boldsymbol{\tau}$ are sublattice pseudo-spin Pauli matrices, $\boldsymbol{\sigma}$ are real spin Pauli matrices, $\Delta_Z = \frac{g}{2}\mu_B B$ is the Zeeman energy ($\mu_B = \frac{e\hbar}{2m_e} = \frac{\pi}{m_e}$ is the Bohr magneton), the magnetic field $B > 0$ is assumed to be along the $z$ direction perpendicular to the conduction plane and $\Delta$ is a staggered on-site energy. In the following we take units such that $\nu = 1$. The LLs are

$$\epsilon_{n,\xi,\lambda,\sigma} = \epsilon_{n,\xi,\lambda} + \epsilon_\sigma = \lambda\sqrt{\Delta^2 + \omega_\nu^2\bar{n}} + \sigma\Delta_Z, \tag{11}$$

where $\lambda = \pm 1$ is the band index, $\sigma = \pm 1$ is the spin index, $\bar{n} = n + \frac{1-\lambda\xi}{2}$ ($n \in \mathbb{N}$ such that $\bar{n} \in \mathbb{N}$ if $\lambda\xi = +$, $\bar{n} \in \mathbb{N}^*$ if $\lambda\xi = -$) and $\omega_\nu \equiv \sqrt{4\pi B}$. It is important to treat correctly the $n = 0$ LL in order to account for the parity anomaly. Therefore:

$$n = \frac{\epsilon^2 - \Delta^2}{\omega_\nu^2} - \frac{1 - \lambda\xi}{2} - \frac{2\Delta_Z\epsilon}{\omega_\nu^2}\sigma + \frac{\Delta_Z^2}{\omega_\nu^2} = \frac{c_0}{B} + c_1 + c_2 B. \tag{12}$$

Here the comparison with equation (9) gives

$$
\begin{aligned}
N_{0,\xi,\sigma}(\epsilon) &= \frac{\epsilon^2 - \Delta^2}{4\pi}\lambda\Theta &\longrightarrow\quad \rho_{0,\xi,\sigma}(\epsilon) = \frac{|\epsilon|}{2\pi}\Theta, \\
M'_{0,\xi,\sigma}(\epsilon) &= -\frac{g}{2}\mu_B\frac{|\epsilon|}{2\pi}\sigma\Theta + \frac{\xi}{2}\Theta = M'_{\text{spin}} + M'_{\text{orb}}, \\
\chi'_{0,\xi,\sigma}(\epsilon) &= \frac{(\frac{g}{2}\mu_B)^2}{2\pi}\lambda\Theta, \\
R'_{p,\xi,\sigma} &= 0 \text{ for } p \geq 3,
\end{aligned}
\tag{13}
$$

where the step function $\Theta \equiv \Theta[|\epsilon| - \Delta]$ is defined in order to avoid cluttered notation and here $\lambda = \text{sign}(\epsilon)$. These quantities are plotted in Figure 1, together with the zero-field energy spectrum and the LLs. The DoS agrees with that found by Koshino and Ando [14]. TRS implies that when summing over spin and valley indices, the spontaneous magnetization should vanish and therefore $M_{0,\xi,\sigma}(\epsilon) = \xi\frac{\epsilon}{2}\Theta - \frac{g}{2}\mu_B\frac{\epsilon^2\lambda}{4\pi}\sigma\Theta$ so that $M_0(\epsilon) = \sum_\xi \sum_\sigma M_{0,\xi,\sigma} = 0$. That $M'_{\text{orb}} = \frac{\xi}{2}$ comes from the peculiar behavior of the winding number $W = \lambda\xi$ of gapped graphene, which leads to a phase shift $\gamma_0$ (see section 5 below) which is energy and magnetic field independent, $\gamma_0 = \frac{1}{2} - M'_{0,\xi} = \frac{1}{2} - \frac{W}{2}$ [15]. The quantization $\gamma_0 = 0$ mod. 1 is actually only exact for the

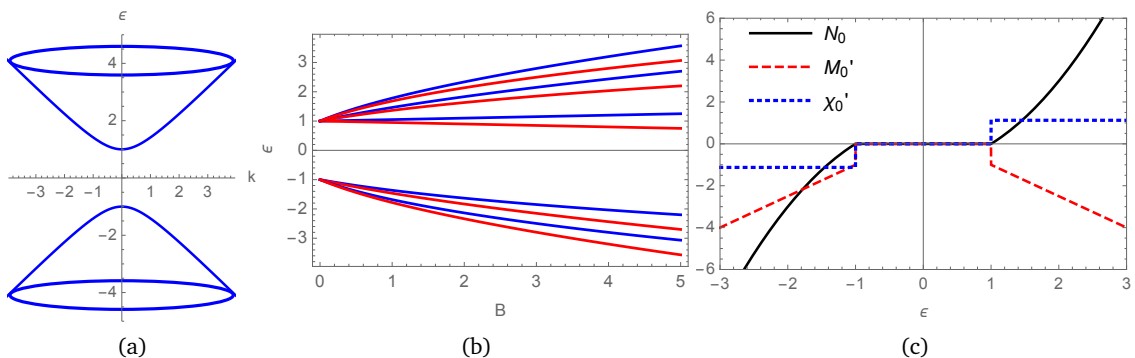

Figure 1: Gapped graphene monolayer. (a) Dispersion relation in zero field: energy $\epsilon_\lambda(k) = \lambda\sqrt{k^2 + \Delta^2}$ [in units of $\Delta$] as a function of the wavevector $k$ [in units of $\Delta$]. (b) Landau levels: energy $\epsilon_{n,\xi,\lambda,\sigma}$ [in units of $\Delta$] as a function of the field $B$ [in units of $\frac{\Delta^2}{4\pi}$] for different values of the Landau index $\bar{n} = n + \frac{1-\lambda\xi}{2} = 0,1,2$ for $\lambda = +$ and $\bar{n} = 1,2$ for $\lambda = -$ (here we consider the $\xi = +$ valley) and a $g$-factor such that the Zeeman energy $\Delta_Z = 0.05\frac{\omega_v^2}{\Delta}$. Blue lines are for spin up and and red line for spin down. (c) Integrated density of states (IDoS) $N_{0,\xi,\sigma}$ [black line, in units of $\frac{\Delta^2}{4\pi}$], derivative of the magnetization $M'_{0,\xi,\sigma}$ [red dashed line] and derivative of the susceptibility $\chi'_{0,\xi,\sigma}$ [dotted blue line, in units of $\frac{4\pi}{\Delta^2}$] as a function of energy $\epsilon$ [in units of $\Delta$] for $\xi = +$, $\sigma = +$, $g = 2$ and $\frac{\Delta}{m_e} = 3$.

simplified two-band model of gapped graphene that we consider here, see [4,16,17] for a discussion. After integrating $\chi'_0$, one recognizes two contributions, the Pauli spin paramagnetism $\chi_{\text{spin}}(\epsilon) = (\frac{g}{2}\mu_B)^2\rho_{0,\xi,\sigma}(\epsilon)$ and a flat orbital susceptibility $\chi_{\text{orb}}(\epsilon) = \text{const}$, which cannot be obtained from this method. Actually, Koshino and Ando find that the orbital susceptibility is piecewise constant $\chi_{\text{orb}}(\epsilon) = -\frac{\pi}{3\Delta}\Theta(\Delta - |\epsilon|)$ [14] rather than constant. This means that the derivative is singular at the gap edge: $\chi'_{\text{orb}}(\epsilon) = \frac{\pi}{3\Delta}\lambda\delta(|\epsilon|-\Delta)$. The singularity is not captured by the present approach based on the Roth-Gao-Niu quantization condition. In the gap closing limit $\Delta \to 0$, the Koshino-Ando orbital susceptibility recovers the singular diamagnetic delta peak $\chi_{\text{orb}}(\epsilon) = -\frac{2\pi}{3}\delta(\epsilon)$, first obtained by McClure [13].

Whenever the energy spectrum in the presence of a magnetic field is the sum of an orbital part and of a spin part (i.e. without cross-terms) as in eq. (11), the magnetic response functions are just the sum of an orbital and of a spin response functions. Generally, for systems with TRS, all odd derivatives ($R'_1 = M'_0$, $R'_3$, etc.) are proportional to the valley index $\xi$, such that upon summing over both valleys, they vanish as expected.

## 3.2 Gapped graphene bilayer

The low energy gapped graphene bilayer Hamiltonian is [18]

$$H = -\frac{1}{2m}[(\Pi_x^2 - \Pi_y^2)\tau_x + 2\xi\Pi_x\Pi_y\tau_y] + \Delta\tau_z, \tag{14}$$

where $\xi = \pm$ is the valley index, $m$ is an effective mass and $2\Delta$ is a gap. The exact LLs are [18] ($n \geq 0$)

$$\epsilon_{n,\lambda,\xi}(B) = \lambda\sqrt{\Delta^2 + \bar{n}(\bar{n}-1)\omega_0^2}, \tag{15}$$

where $\omega_0 = \frac{2\pi B}{m}$, $\bar{n} = n + 1 + \lambda\xi$ with $\lambda = \pm$ the band index (the $n = 0$ and $n = 1$ LLs given in [1], which are supposed to be those for $\lambda = +$ and $\xi = +$, are not correct). Inverting this

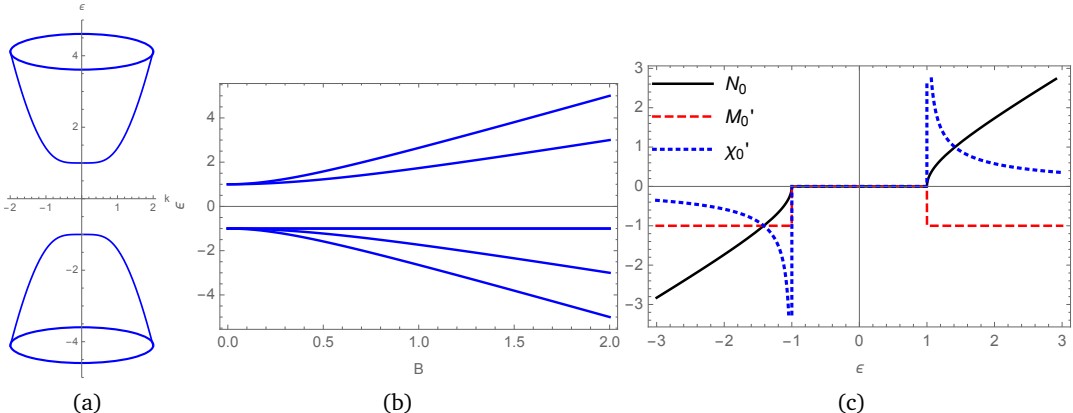

Figure 2: Gapped graphene bilayer. (a) Dispersion relation in zero field: energy $\epsilon_\lambda(k) = \lambda \sqrt{\left(\frac{k^2}{2m}\right)^2 + \Delta^2}$ [in units of $\Delta$] as a function of the wavevector $k$ [in units of $\sqrt{2m\Delta}$]. (b) Landau levels: energy $\epsilon_{n,\xi,\lambda}$ [in units of $\Delta$] as a function of the field $B$ [in units of $\frac{m\Delta}{2\pi}$] for different values of the Landau index $\bar{n} = n + 1 + \lambda\xi = 2, 3$ for $\lambda = +$ and $\bar{n} = 0, 1, 2, 3$ for $\lambda = -$ (here we consider a single valley $\xi = +$). (c) Integrated density of states (IDoS) $N_{0,\xi}$ [black line, in units of $\frac{m\Delta}{2\pi}$], derivative of the magnetization $M'_{0,\xi}$ [red dashed line] and derivative of the susceptibility $\chi'_{0,\xi}$ [blue dotted line, in units of $\frac{\pi}{2m\Delta}$] as a function of energy $\epsilon$ [in units of $\Delta$].

relation we obtain:

$$n = -(\lambda\xi + \frac{1}{2}) + \frac{\sqrt{\epsilon^2 - \Delta^2}}{\omega_0} \sqrt{1 + \frac{\omega_0^2}{4(\epsilon^2 - \Delta^2)}}. \tag{16}$$

Substituting this relation in the l.h.s. of eq.(9) and expanding as a series in powers of $B$, we obtain

$$(n + \frac{1}{2})B = \frac{m}{2\pi}\sqrt{\epsilon^2 - \Delta^2} - \lambda\xi B + \frac{\pi}{4m}\frac{B^2}{\sqrt{\epsilon^2 - \Delta^2}} - \frac{\pi^2}{16m^3}\frac{B^4}{(\epsilon^2 - \Delta^2)^{3/2}} + \cdots, \tag{17}$$

Order by order comparison with the r.h.s. implies that

$$
\begin{aligned}
N_{0,\xi}(\epsilon) &= \frac{m}{2\pi}\sqrt{\epsilon^2 - \Delta^2}\,\lambda\Theta, \\
M'_{0,\xi}(\epsilon) &= -\xi\Theta, \\
\chi'_{0,\xi}(\epsilon) &= \frac{\pi}{2m}\frac{\lambda}{\sqrt{\epsilon^2 - \Delta^2}}\Theta, \\
R'_3(\epsilon) &= 0, \\
R'_4(\epsilon) &= -\frac{3\pi^2}{8m^2}\frac{\lambda}{(\epsilon^2 - \Delta^2)^{3/2}}\Theta,
\end{aligned}
\tag{18}
$$

where as before $\Theta \equiv \Theta[|\epsilon| - \Delta]$ and $\lambda = \text{sign}(\epsilon)$. The above results are plotted in Figure 2. They are compatible with those obtained in [1] except for the sign of $M'_{0,\xi}$.[2] The derivative of the spontaneous magnetization vanishes when summed over both valleys, as it should for

---

[2]The derivative of the spontaneous magnetization is $M'_{0,\xi} = -\xi$ so that it should be $M'_{0,+} = -1$ and not $M'_{0,+} = 1$ for the $\xi = +$ valley. This mistake is related to the wrong LLs that [1] started with. They considered the Hamiltonian for the $\xi = +$ valley. The conduction band LLs should therefore be $\epsilon_{n,\lambda=+,\xi=+} = \sqrt{\Delta^2 + \omega_0^2 n(n-1)}$ for $n \geq 2$ and $\epsilon_{n,\lambda=+,\xi=+} = -\Delta$ (and not $+\Delta$) for $n = 0$ and 1.

a system with TRS. Also $M'_{0,\xi} = -\xi$ corresponds to a winding number $W = 2\lambda\xi$ as shown in [15] through the relation $\gamma_0 = \frac{1}{2} - \frac{W}{2}$ for the phase shift (see section 5 below). The result for the derivative of the susceptibility agrees with the response function found by Safran [19] for a gapless $\Delta \to 0$ bilayer (per valley and per spin projection)

$$\chi_0(\epsilon) = \frac{\pi}{2m}\left(\frac{1}{3} + \ln\frac{|\epsilon|}{t_\perp}\right) \quad \longrightarrow \quad \chi'_0 = \frac{\pi}{2m}\frac{1}{\epsilon}, \tag{19}$$

where $t_\perp$ is a high-energy cutoff related to interlayer hoping (see also [21, 22]).

We conclude this section by discussing a related but different case, namely that of a quadratic band crossing point that occurs at a time-reversal invariant momentum, such as the M point of the checkerboard (Mielke) lattice [23]. The low-energy Hamiltonian reads

$$H = \frac{1}{2m}[(\Pi_x^2 - \Pi_y^2)\tau_z + 2\Pi_x\Pi_y\tau_x]. \tag{20}$$

The main differences with the gapless graphene bilayer Hamiltonian, eq. (14) with $\Delta = 0$, are in the involved Pauli matrices, which reflect time-reversal invariance and the fact that there is a single valley. In this situation, the Landau levels are

$$\epsilon_{n,\lambda}(B) = \lambda\omega_0\sqrt{n(n+1)}, \tag{21}$$

where $n = 0, 1, 2...$ and $\lambda = \pm$. Note that the change in Pauli matrices does not affect the energy spectrum but it affects the labelling (i.e. the quantum numbers $n$ and $\lambda$), which matters to deduce the response functions. Inverting this equation, we find

$$(n + \frac{1}{2})B = \frac{m\epsilon}{2\pi}\sqrt{1 + \frac{\omega_0^2}{4\epsilon^2}} \approx \frac{m\epsilon}{2\pi} + \frac{\pi B^2}{4m\epsilon} + \dots, \tag{22}$$

which, in contrast to eq. (16) does not contain the valley term $\lambda\xi$. Then we deduce :

$$\begin{aligned} N_0(\epsilon) &= \frac{m\epsilon}{2\pi}, \\ M'_0(\epsilon) &= 0, \\ \chi'_0(\epsilon) &= \frac{\pi}{2m\epsilon}. \end{aligned} \tag{23}$$

We then correctly find that the magnetization (actually its derivative) is zero in this case, while it was finite and of opposite signs in the previous case with two valleys. In both cases the total magnetization is zero due to TRS.

### 3.3 Two-dimensional semiconductors with gapped Dirac electrons

The low energy Hamiltonian of gapped Dirac electrons in two-dimensional semiconductors such as transition metal dichalcogenides (see e.g. [16]) is

$$H_\xi = \frac{\Pi_x^2 + \Pi_y^2}{2m}\mathbb{1} + \nu(\xi\Pi_x\tau_x + \Pi_y\tau_y) + (\Delta + \frac{\Pi_x^2 + \Pi_y^2}{2m_z})\tau_z, \tag{24}$$

where $\xi = \pm 1$ is the valley index and $(\tau_x, \tau_y, \tau_z)$ are pseudo-spin Pauli matrices. In the following, we take units such that $\nu = 1$. The exact LLs are given by (for $n \geq 0$):

$$\epsilon_{n,\lambda,\xi} = \omega_0\bar{n} - \omega_z\frac{\xi}{2} + \lambda\sqrt{\left(\Delta + \omega_z\bar{n} - \omega_0\frac{\xi}{2}\right)^2 + \omega_\nu^2\bar{n}}, \tag{25}$$

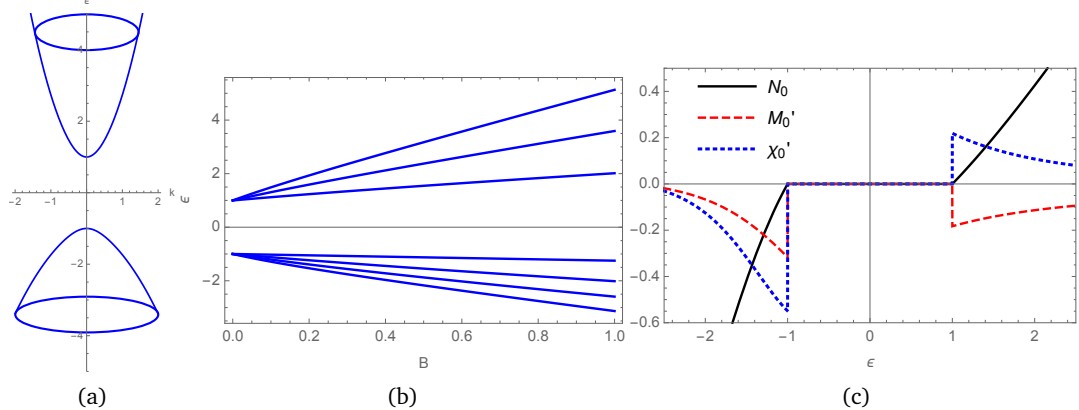

Figure 3: 2D semiconductors. (a) Dispersion relation in zero field: energy $\epsilon_\lambda(k) = \frac{k^2}{2m} + \lambda\sqrt{k^2 + \left(\Delta + \frac{k^2}{2m_z}\right)^2}$ [in units of $\Delta$] as a function of the wavevector $k$ [in units of $\Delta$] for $\frac{\Delta}{2m_z v^2} = 1 > \frac{\Delta}{2m} = 0.5$. (b) Landau levels: energy $\epsilon_{n,\xi,\lambda}$ [in units of $\Delta$] as a function of the field $B$ [in units of $\frac{\Delta^2}{4\pi}$] for different values of the Landau index $\bar{n} = n + \frac{1+\lambda\xi}{2} = 1,2,3$ for $\lambda = +$ and $\bar{n} = 0,1,2,3$ for $\lambda = -$ (here we consider the $\xi = +$ valley). (c) IDoS $N_0$ [black line, in units of $\frac{\bar{m}\Delta}{2\pi}$], derivative of the magnetization $M_0'$ [red dashed line] and derivative of the susceptibility $\chi_0'$ [dotted blue line, in units of $\frac{\pi}{2\bar{m}\Delta}$] as a function of the energy $\epsilon$ [in units of $\Delta$] for $\frac{\bar{m}}{m_z} = \frac{2}{\sqrt{3}}$, $\frac{\bar{m}}{m} = \frac{1}{\sqrt{3}}$ and $\frac{\bar{m}}{\Delta} = 1$.

where $\omega_0 = \frac{2\pi B}{m}$, $\omega_z = \frac{2\pi B}{m_z}$, $\omega_v = \sqrt{4\pi B}$, $\bar{n} = n + \frac{1+\lambda\xi}{2}$ and with $\lambda = \pm 1$ the band index.

Inverting this relation, we obtain

$$\bar{n} = \frac{\bar{m}^2}{2\pi B}\left[-\left(\frac{\bar{\Delta}}{m_z} + \frac{\bar{\epsilon}}{m} + 1\right) + \sqrt{\left(\frac{\bar{\Delta}}{m_z} + \frac{\bar{\epsilon}}{m} + 1\right)^2 + \frac{1}{\bar{m}^2}(\bar{\epsilon}^2 - \bar{\Delta}^2)}\right], \quad (26)$$

where we have defined $\frac{1}{\bar{m}^2} = \frac{1}{m_z^2} - \frac{1}{m^2} > 0$, $\bar{\Delta} = \Delta - \frac{\xi\pi B}{m}$ and $\bar{\epsilon} = \epsilon + \frac{\xi\pi B}{m_z}$. The assumption that $\frac{1}{m_z} > \frac{1}{m}$ insures that the Fermi surface is made of a single piece.

Using relation eq. (9), and expanding to order $B^2$ we obtain successively:

$$N_0 = \frac{\bar{m}^2}{2\pi}\left[-\left(\frac{\Delta}{m_z} + \frac{\epsilon}{m} + 1\right) + \sqrt{\left(\frac{\Delta}{m_z} + \frac{\epsilon}{m} + 1\right)^2 + \frac{1}{\bar{m}^2}(\epsilon^2 - \Delta^2)}\right]\lambda\Theta, \quad (27)$$

$$M_0' = \frac{\xi}{2}\left[-1 + \frac{\left(\frac{\Delta}{m} + \frac{\epsilon}{m_z}\right)\lambda}{\sqrt{\left(\frac{\Delta}{m_z} + \frac{\epsilon}{m} + 1\right)^2 + \frac{1}{\bar{m}^2}(\epsilon^2 - \Delta^2)}}\right]\Theta, \quad (28)$$

and

$$\chi_0' = \frac{\pi}{2\bar{m}^2}\frac{\lambda\Theta}{\sqrt{\left(\frac{\Delta}{m_z} + \frac{\epsilon}{m} + 1\right)^2 + \frac{1}{\bar{m}^2}(\epsilon^2 - \Delta^2)}}\left[1 - \frac{\left(\frac{\Delta}{m} + \frac{\epsilon}{m_z}\right)^2}{\left(\frac{\Delta}{m_z} + \frac{\epsilon}{m} + 1\right)^2 + \frac{1}{\bar{m}^2}(\epsilon^2 - \Delta^2)}\right] \quad (29)$$

where $\Theta \equiv \Theta[|\epsilon| - \Delta]$ and $\lambda = \text{sign}(\epsilon)$. These results are plotted in Figure 3. Despite a complicated expression, they correspond to simple functions. The derivative of the magnetization vanishes when summed over both valleys as expected for a system with TRS. This is actually the case of all odd response functions.

Equation (28) shows that, for this 2D semiconductor model, the valley magnetization contribution is energy dependent and not quantized, in contrast to gapped Dirac electrons (see

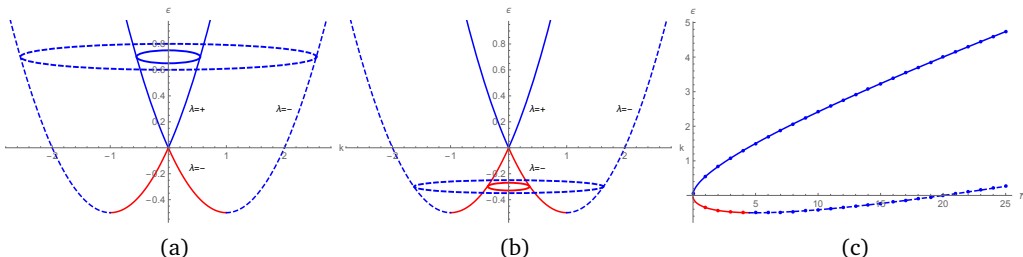

Figure 4: Rashba model. (a) and (b): Energy spectrum $\epsilon_\lambda(k) = \frac{k^2}{2m} + \lambda k$ [$\epsilon$ is in units of $m$] as a function of the wavevector $k$ [in units of $m$]. The dispersion relation is isotropic in $(k_x, k_y)$ space. The inner branches are plotted in full lines and the outer branches in dashed lines. The electron-like branches are in blue and the hole-like branches are in red. Fermi circles are also indicated for the case of positive (a) or negative energy (b). (c) LLs: energy $\epsilon_{\bar{n},\lambda}$ [in units of $m$] as a function of the Landau index $\bar{n} = n + \frac{1-\lambda}{2}$ between 0 and 25 for $\lambda = +$ (blue) and between 1 and 25 for $\lambda = -$ (red for hole-like and dashed blue for electron-like LLs). The magnetic field is fixed such that $\frac{2\pi B}{m^2} = 0.1$ and we considered the case $\tilde{g} = 0$.

section 3.1). As a consequence, it would contribute an energy-dependent phase shift in magnetic oscillations (see section 5 and [4,17]).

## 3.4 Rashba model

As a last example, we consider the case of a two-dimensional electron gas with Rashba spin-orbit coupling and a Zeeman effect, known as the Rashba model [24]. In contrast to the previous examples, the spectrum of this model is gapless. The Hamiltonian reads [24]

$$H = \frac{\Pi^2}{2m}\sigma_0 + v(\Pi_y\sigma_x - \Pi_x\sigma_y) + \Delta_Z\sigma_z, \tag{30}$$

where $m$ is the mass, $v$ the velocity (i.e. the Rashba parameter quantifying the spin-orbit coupling), $\Delta_Z$ the Zeeman energy and the magnetic field $B > 0$ is assumed to be along the $z$ direction perpendicular to the conduction plane. The Pauli matrices $(\sigma_x, \sigma_y, \sigma_z)$ describe the true spin and there is a single valley (e.g. at the A point of the hexagonal Brillouin zone in the case of BiTeI, see [25]). This is also an effective description of the surface states of three-dimensional topological insulators [4, 26, 27]. However, there is a subtle distinction between the case of the Rashba model and that of the topological insulator: in the first case, the Fermi surface is made of two disconnected pieces because the parabolic term in the Hamiltonian is not a small correction to the linear term, in contrast to the second case in which there is a single Fermi surface. The correct treatment of the Rashba model therefore requires great care. In the following we take units such that $v = 1$ in addition to $\hbar = 1$ and $e = 2\pi$.

The dispersion relation is made of two bands $\epsilon_\lambda(k) = \frac{k^2}{2m} + \lambda k$ where $\lambda = \pm$ is a band index (note that here $\lambda$ is not the same as the sign of the energy) and $k = |\mathbf{k}|$. The Fermi surface is made of two pieces: an inner circle with radius $k_i$ and an outer circle with radius $k_o$ (see Figure 4). At positive energy, the inner circle belongs to the $\lambda = +$ band ($k_i = -m + m\sqrt{1 + \frac{2\epsilon}{m}}$) and the outer circle to the $\lambda = -$ band ($k_o = m + m\sqrt{1 + \frac{2\epsilon}{m}}$). Both circles are electron-like. At negative energy ($-\frac{m}{2} < \epsilon < 0$), the Fermi surface is again made of two circles but both the inner and the outer circles belong to the $\lambda = -$ band ($k_i = m - m\sqrt{1 - \frac{2|\epsilon|}{m}}$ and $k_o = m + m\sqrt{1 - \frac{2|\epsilon|}{m}}$) and the $\lambda = +$ band no longer intersects the Fermi energy. The inner circle is hole-like and the outer circle is electron-like. For future need, we introduce a branch index $b = o/i = \pm$ to

designate the two Fermi circles: $o$ for outer $(+)$ and $i$ for inner $(-)$. Therefore

$$k_b = m \left| 1 + b \sqrt{1 + \frac{2\epsilon}{m}} \right|, \qquad (31)$$

which is valid for $b = \pm$ and for positive or negative $\epsilon$. In the case of the Rashba model, it is essential to consider the two Fermi circles. However, when describing surface states of topological insulators, the Rashba model is only valid for small wave-vectors and therefore only the inner circle should be considered (in Figure 4(a) or (b), it means that only the full line should be considered and not the dashed one). The latter model was already discussed by Gao and Niu [1]. It leads to inconsistencies such as a non-vanishing magnetization despite TRS.

The LLs are [24,28]

$$\epsilon_{\bar{n},\lambda} = \omega_0 \bar{n} + \lambda \sqrt{\omega_v^2 \bar{n} + \frac{\omega_0^2}{4}(1 - \tilde{g})^2}, \qquad (32)$$

where $\omega_0 = \frac{2\pi B}{m}$, $\omega_v = \sqrt{4\pi B}$, $\tilde{g} = \frac{2\Delta_Z}{\omega_0} = \frac{g}{2}\frac{m}{m_e}$ and we assume that $\tilde{g} < 1$. In particular, $\bar{n} = n + \frac{1-\lambda}{2}$ with $n = 0, 1, 2, ...$, which means that $\bar{n} \in \mathbb{N}$ when $\lambda = +$ and $\bar{n} \in \mathbb{N}^*$ when $\lambda = -$ so that the parity anomaly at $n = 0$ is correctly taken into account. Indeed at $\bar{n} = 0$ there is a single LL ($\lambda = +$ with our choice), whereas for any $\bar{n} > 1$, there are always two LLs (one with $\lambda = +$ and one with $\lambda = -$). It will be important to keep in mind that some of these LLs are electron-like (inner and outer circles at positive energy and outer circle at negative energy) and some are hole-like (inner circle at negative energy), see Figure 4(c) and 5(a).

Inverting this equation with care, we find

$$n_b = \frac{c_0 + b\sqrt{c_1 + c_2 B^2}}{B} - \frac{1}{2} - \frac{b\Theta(\epsilon) + \Theta(-\epsilon)}{2}, \qquad (33)$$

with $c_0 = \frac{m^2}{2\pi}(1 + \frac{\epsilon}{m})$, $c_1 = \frac{m^4}{4\pi^2}(1 + \frac{2\epsilon}{m})$ and $c_2 = \frac{(1-\tilde{g})^2}{4}$. Upon expansion in powers of $B$, this gives

$$n_b = \frac{c_0 + b\sqrt{c_1}}{B} - \frac{1}{2} - \frac{b\Theta(\epsilon) + \Theta(-\epsilon)}{2} + b\frac{c_2}{\sqrt{c_1}}\frac{B}{2} + \mathcal{O}(B^3). \qquad (34)$$

Equation (9) was obtained for electron-like LLs and is therefore valid only for the outer circle (for all energies) and for the inner circle at positive energy. However, we need to modify it for the hole-like LLs corresponding to the inner circle at negative energy. In that case, see equation (8), it becomes

$$n = \frac{N_{\text{tot}} - N(\epsilon, B)}{B} - \frac{1}{2} \approx \frac{N_{\text{tot}} - N_0(\epsilon)}{B} - \frac{1}{2} - M_0'(\epsilon) - \frac{B}{2}\chi_0'(\epsilon), \qquad (35)$$

where $N_{\text{tot}}$ is a constant.

By comparing eq. (34) with eq. (9) and (35), we obtain the IDoS, the magnetization and the susceptibility. We start with the IDoS. For $\epsilon > 0$, the IDoS is $N_{0,b}(\epsilon) = \frac{m^2}{2\pi}(1 + \frac{\epsilon}{m} + b\sqrt{1 + \frac{2\epsilon}{m}}) = \frac{\pi k_b^2}{(2\pi)^2}$ so that $N_0 = \sum_b N_{0,b} = \frac{m^2}{2\pi}(1 + \frac{\epsilon}{m})$ and the DoS is $\rho_0(\epsilon) = \frac{m}{\pi}$ as expected. For $-\frac{m}{2} < \epsilon < 0$, we find that $N_{0,b}(\epsilon) = \frac{m^2}{2\pi}(b + b\frac{\epsilon}{m} + \sqrt{1 + \frac{2\epsilon}{m}})$ so that $N_0 = \frac{m^2}{\pi}\sqrt{1 + \frac{2\epsilon}{m}}$ and the DoS is $\rho_0(\epsilon) = \frac{m}{\pi}\frac{1}{\sqrt{1 + 2\epsilon/m}}$ featuring a 1D-like van Hove singularity when $\epsilon \to -\frac{m}{2}$ corresponding to the minimum of the mexican hat dispersion relation. In (35), we took $N_{\text{tot}} = 0$ in order that the total IDoS be continuous at $\epsilon = 0$. The IDoS agrees

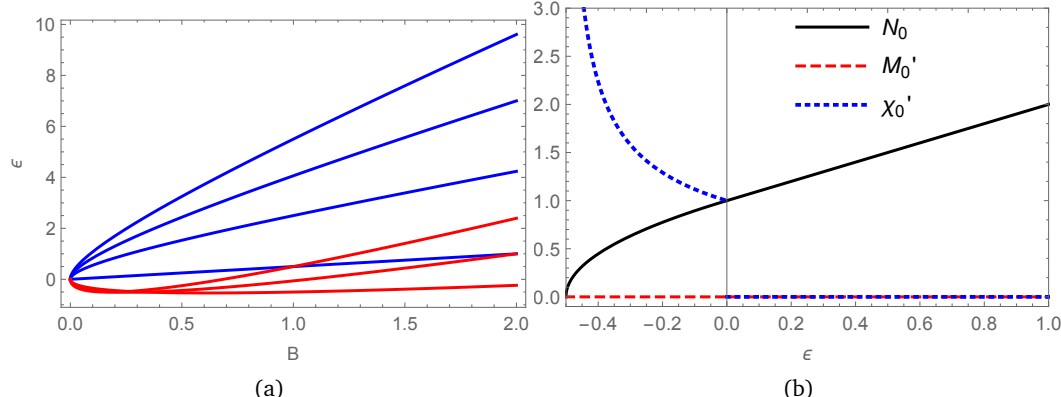

Figure 5: Rashba model. (a) Landau levels $\epsilon_{\bar{n},\lambda}$ [in units of $m$] as a function of the magnetic field $B$ [in units of $\frac{m^2}{2\pi}$] for $\bar{n} = 0, 1, 2, 3$ when $\lambda = +$ [in blue] and for $\bar{n} = 1, 2, 3$ when $\lambda = -$ [in red]. (b) Integrated density of states (IDoS) $N_0$ [black full line, in units of $\frac{m^2}{\pi}$], derivative of the magnetization $M_0' = 0$ [red dashed line] and derivative of the susceptibility $\chi_0'$ [blue dotted line, in units of $\frac{\pi}{m^2}$ and for $\tilde{g} = 0$] as a function of the energy $\epsilon$ [in units of $m$]. Compare with [28].

with $N_0 = \frac{\pi k_o^2 - \pi k_i^2}{(2\pi)^2}$. The change of sign in front of $k_i^2$ is due to the fact that the inner Fermi circle is hole-like. In the end, restoring the units, the IDoS is

$$N_0(\epsilon) = \frac{m^2 v^2}{\pi \hbar^2}\left[\sqrt{1 + \frac{2\epsilon}{mv^2}}\Theta(-\epsilon)\Theta(\epsilon + \frac{mv^2}{2}) + \left(1 + \frac{\epsilon}{mv^2}\right)\Theta(\epsilon)\right], \qquad (36)$$

which tends to $\frac{m\epsilon}{\pi \hbar^2}\Theta(\epsilon)$ in the $v \to 0$ limit as expected (see Figure 5).

The derivative of the magnetization is $M_{0,b}' = -\frac{b}{2}$ so that $M_0' = \sum_b M_{0,b}' = 0$ and the total magnetization $M_0(\epsilon) = $ const as a consequence of the compensation between the inner and outer Fermi circles. TRS actually implies that $M_0(\epsilon) = 0$.

We obtain the derivative of the susceptibility as:

$$
\begin{aligned}
\chi_{0,b}'(\epsilon) &= \frac{\pi}{2m^2}\frac{(1-\tilde{g})^2}{\sqrt{1 + \frac{2\epsilon}{m}}} \quad \text{when } \epsilon < 0 \\
&= b\frac{\pi}{2m^2}\frac{(1-\tilde{g})^2}{\sqrt{1 + \frac{2\epsilon}{m}}} \quad \text{when } \epsilon > 0.
\end{aligned}
\qquad (37)
$$

For positive energy and the inner circle $b = i = -$, this result agrees with [1] (see in particular the supporting information of this article). The total (derivative of the) susceptibility is therefore

$$
\begin{aligned}
\chi_0'(\epsilon) &= \frac{\pi}{m^2}\frac{(1-\tilde{g})^2}{\sqrt{1 + \frac{2\epsilon}{m}}} \quad \text{when } \epsilon < 0 \\
&= 0 \quad \text{when } \epsilon > 0,
\end{aligned}
\qquad (38)
$$

and is displayed in Figure 5. Except for a delta-like singularity at $\epsilon = 0$, which is the McClure result mentioned in the section on graphene [13], this compares well with the susceptibility

found by Suzuura and Ando [28]

$$
\begin{aligned}
\chi_0(\epsilon) &= \frac{\pi}{m}(1-\tilde{g})^2\sqrt{1+\frac{2\epsilon}{m}} \text{ when } -\frac{m}{2} < \epsilon < 0, \\
&= -\frac{2\pi}{3}\delta(\epsilon) \text{ when } \epsilon \approx 0, \\
&= \frac{\pi}{m}(\tilde{g}^2 - \frac{1}{3}) \text{ when } \epsilon > 0,
\end{aligned}
\tag{39}
$$

the derivative of which is:

$$
\begin{aligned}
\chi_0'(\epsilon) &= \frac{\pi}{m^2}\frac{(1-\tilde{g})^2}{\sqrt{1+\frac{2\epsilon}{m}}} \text{ when } -\frac{m}{2} < \epsilon < 0, \\
&= -\frac{2\pi}{3}\delta'(\epsilon) \text{ when } \epsilon \approx 0, \\
&= 0 \text{ when } \epsilon > 0.
\end{aligned}
\tag{40}
$$

However it slightly disagrees with [25]. In the vicinity of $\epsilon = 0$, these authors find a diamagnetic peak $\propto -\frac{1}{|\epsilon|}$, instead of the expected McClure [13] delta singularity for a Dirac cone.[3]

## 4 From magnetic response functions to Landau levels

One may also use the Roth-Gao-Niu quantization condition in order to obtain an approximate analytical expression of LLs. As an input, one needs zero-field quantities such as the IDoS $N_0(\epsilon)$, the spontaneous magnetization $M_0(\epsilon)$ and the magnetic susceptibility $\chi_0(\epsilon)$.

The general structure of LLs obtained by including successive powers of $B$ in the Roth-Gao-Niu relation (7) can be shown to be

$$
\epsilon_n(B) = \sum_{p \geq 0} g_p(x)B^p = \sum_{p \geq 0} g_p(x)x^p \frac{1}{(n+\frac{1}{2})^p},
\tag{41}
$$

where $x \equiv (n+\frac{1}{2})B$ and the functions $g_p(x)$ depend on the magnetic response functions. At zeroth order in $B$, $g_0(x) = N_0^{-1}(x) = \epsilon_n^{(0)}$, which is the Onsager result. At first order,

$$
g_1(x) = -\frac{M_0'(\epsilon_n^{(0)})}{N_0'(\epsilon_n^{(0)})} = -\frac{M_0'}{N_0'},
\tag{42}
$$

and at second order:

$$
g_2(x) = \frac{M_0'M_0''}{N_0'^2} - \frac{M_0'^2 N_0''}{2N_0'^3} - \frac{\chi_0'}{2N_0'}.
\tag{43}
$$

At $p$th order, the deviation with respect to the exact LLs is expected to be of order $B^{p+1}$ (at fixed $x$).

Below, we illustrate this idea on three examples: (1) the square lattice tight-binding model (Hofstadter problem), (2) the semi-Dirac Hamiltonian and (3) the gapped graphene bilayer.

---

[3]There is also a numerical mistake in equation (4) of [25]: the prefactor should be $\frac{1}{4\pi}$ instead of $\frac{5}{32\pi}$.

## 4.1 Square lattice tight-binding model

We consider the square lattice tight-binding model in a perpendicular magnetic field (the famous Hofstadter model [29]) and choose units such that $a = 1$ (and not $\mathscr{A} = 1$ here), $t = 1$ and $\phi_0 = 1$ ($\hbar = 1$ and $e = 2\pi$) so that the flux per plaquette is $f = Ba^2/\phi_0 = B$. The idea is to use the Roth-Gao-Niu relation to reproduce features of the Hofstadter spectrum. This example is particularly interesting since the quantities entering the Roth-Gao-Niu relation are known *analytically*. In particular, we recover a semi-classical expression of the LLs up to third order in the magnetic flux, originally obtained by Rammal and Bellissard [30].

As the magnetization vanishes, the Roth-Gao-Niu relation reads:

$$(n + \frac{1}{2})f \simeq N_0(\epsilon) + \frac{f^2}{2}\chi_0'(\epsilon). \tag{44}$$

Time-reversal symmetry implies that the expansion in powers of $f$ of the grand potential only contains even powers. The above expression is therefore valid up to order $f^4$.
The zero-field DoS $\rho_0(\epsilon) = N_0'(\epsilon)$ is [31]

$$\rho_0(\epsilon) = \frac{1}{2\pi^2}K(\sqrt{1 - \epsilon^2/16}) , \tag{45}$$

where $K(x)$ is the complete elliptic integral of the first kind [32].[4] The susceptibility $\chi_0$ has been also calculated and can be written in the form [22,33]:

$$\chi_0(\epsilon) = -\frac{\pi}{6}\left[(\epsilon^2 - 8)\rho_0(\epsilon) - E(\epsilon)\right]$$
$$= \frac{2}{3}Q_{1/2}(1 - \epsilon^2/8) . \tag{46}$$

$E(\epsilon) = \int_0^\epsilon v\rho_0(v)dv$ is the total energy and $Q_\nu(z)$ is a Legendre function of the second kind [32]. The derivative $\chi_0'(\epsilon)$ is

$$\chi_0'(\epsilon) = -\frac{\pi}{6}\left[(\epsilon^2 - 8)\rho_0'(\epsilon) + \epsilon\rho_0(\epsilon)\right] . \tag{47}$$

Inserting in Eq.(44) the expressions of the IDoS $N_0(\epsilon)$ obtained from integrating Eq.(45) and of the derivative $\chi_0'(\epsilon)$, we obtain the position $\epsilon_n(f)$ of the LLs in the inverted form:

$$f(\epsilon_n) = \frac{(n + 1/2) - \sqrt{(n + 1/2)^2 - 2N_0(\epsilon_n)\chi_0'(\epsilon_n)}}{\chi_0'(\epsilon_n)} . \tag{48}$$

One recovers the Onsager quantization rule in the limit of a constant susceptibility ($\chi_0' \to 0$). By solving this equation, one obtains the field dependence of the LLs $\epsilon_n(f)$. They are plotted in Fig. 6 and are compared with the exact levels of the Hofstadter spectrum. It provides a very good description of the global evolution of the Hofstadter bands. More quantitatively, Fig. 7 compares the obtained $\epsilon_n(f)$ with the average energy of the Landau-Hofstadter bands. On the same figure, we also plot the position of the middle of the Landau bands for values of the flux $f$ of the form $f = 1/q$ that is when the Landau band consists in a single sub-band (the middle then corresponds to the position of the Van Hove singularity).

As shown in Fig. 6, this excellent fit works as long as a semi-classical quantization relation is applicable, that is as long as the number of states in a broadened Landau level equals the Landau degeneracy $f\mathscr{A}$. When $f_{\max}(n) = \frac{1}{2(n+1)}$, the broadened level $n$ overlaps with its symmetric in energy (coming from the top of the band) and the broadened level begins to empty instead of following the Landau degeneracy.

---

[4] We use the definition of the elliptic integrals from Gradshteyn and Ryzhik [32]. They differ from those used in Mathematica: $K_{Grad.}(x) = K_{Math.}(x^2)$.

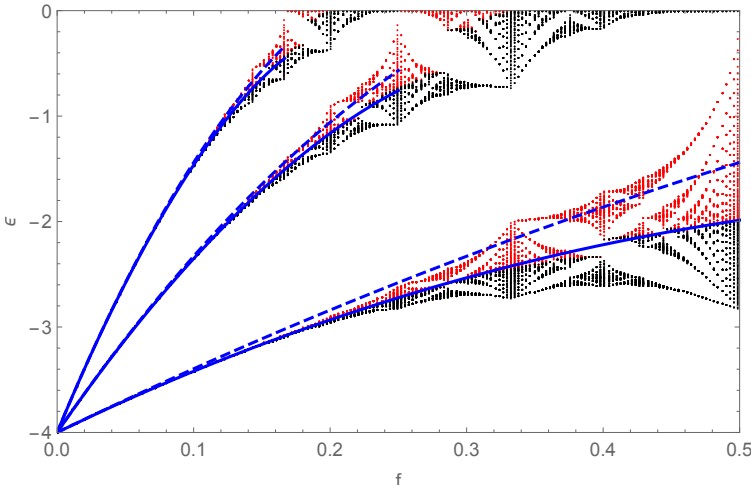

Figure 6: Bottom part of Hofstadter spectrum (energy as a function of the flux per plaquette). The dashed blue lines are the three first LLs ($n = 0, 1, 2$) obtained from the usual Onsager semi-classical quantization rule with the exact zero-field IDoS. The blue lines are obtained from the Roth-Gao-Niu rule with the susceptibility correction Eq.(48). Semi-classical LLs are plotted up to a maximum flux $f_{\max} = \frac{1}{2(n+1)}$. The color code for the Hofstadter energy levels obtained for rational values of $f = p/q$ with $q = 349$ separate the lower half of the energy levels (black) from the upper half (red).

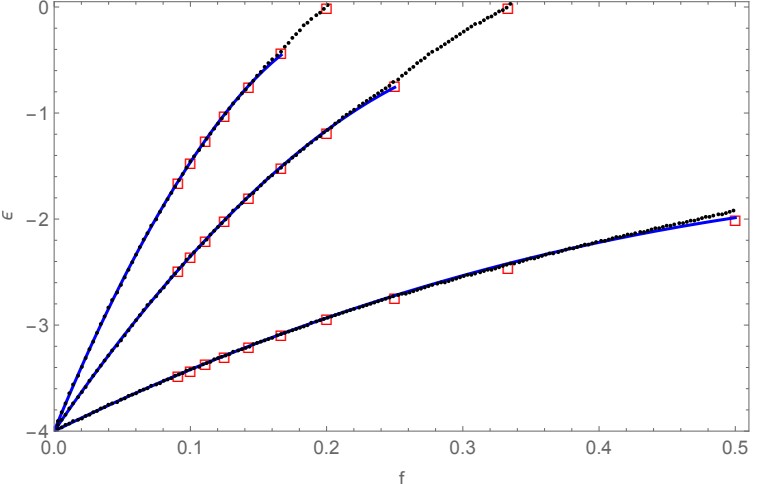

Figure 7: The position of the LLs $\epsilon_n^{\text{GN}}(f)$ obtained from the Roth-Gao-Niu rule (blue line) fits very well the field dependence of the average energy of the Landau bands (black dots) obtained from the Hofstadter spectrum. The position of the middle of the bands when $f = 1/q$ is also plotted (red squares). These fits work perfectly well up to the maximal flux $f_{\max}(n) = \frac{1}{2(n+1)}$ (where $n = 0, 1, 2, \cdots$ is the Landau index) and start to deviate beyond that.

From the implicit Eq. (48), one can easily obtain a series expansion of the field dependence $\epsilon_n(f)$. From Eqs.(45,46), we obtain the following expansions, defining $\mu = \epsilon + 4$

$$\rho_0(\epsilon) = \frac{1}{4\pi} + \frac{1}{32\pi}\mu + \frac{5}{1024\pi}\mu^2 + \cdots, \tag{49}$$

$$\chi_0(\epsilon) = -\frac{\pi}{3} + \frac{\pi}{8}\mu + \frac{\pi}{256}\mu^2 + \cdots, \tag{50}$$

and by inversion of Eq. (48) order by order in $f$, we obtain

$$\epsilon_n^{\mathrm{GN}}(f) = -4 + 2\pi f(2n+1) - \frac{(2\pi f)^2}{16}[(2n+1)^2 + 1] + \frac{(2\pi f)^3}{192}[n^3 + (n+1)^3] + \dots \quad (51)$$

Remarkably, we recover a result obtained by Rammal and Bellissard (equation (3.28) in [30]) after a lengthy calculation from a totally different method. As we are using a Roth-Gao-Niu relation valid up to order $f^3$ included (but not at order $f^4$ as we do not know the response functions $R_p$ with $p \geq 4$), there is no point in getting the semi-classical LLs beyond that order. Note that using the Onsager relation instead already leads to problems at order $f^2$ for the LLs.

The above LLs can be rewritten in terms of the combinaison $2n+1$ as:

$$\epsilon_n^{\mathrm{GN}}(f) = -4 + 2\pi f(2n+1) - \frac{(2\pi f)^2}{16}[(2n+1)^2 + 1] + \frac{(2\pi f)^3}{192}\frac{(2n+1)^3 + 3(2n+1)}{4} + \dots \quad (52)$$

The Onsager quantization gives the following LLs

$$\epsilon_n^{\mathrm{On}}(f) = -4 + 2\pi f(2n+1) - \frac{(2\pi f)^2}{16}[(2n+1)^2 + 0] + \frac{(2\pi f)^3}{192}\frac{(2n+1)^3 + 0}{4} + \dots, \quad (53)$$

which are a function of $(2n+1)f$ only. In order to compare these two results and their order in the semi-classical expansion, we take $(2n+1)f$ as a constant and count the powers of $f$ (see the corresponding discussion in point (v) of section 2). The comparison of (52) and (53) gives $\epsilon_n^{\mathrm{GN}} - \epsilon_n^{\mathrm{On}} \approx -\frac{(2\pi f)^2}{16} + \frac{(2\pi f)^3}{192}\frac{3(2n+1)}{4} \propto f^2$. The Onsager LLs are $\epsilon_n^{\mathrm{On}} \propto f^0 + \mathcal{O}(f^2)$, whereas the Roth-Gao-Niu LLs are $\epsilon_n^{\mathrm{GN}} \propto f^0 + f^2 + \mathcal{O}(f^4)$. The $f^1$ and $f^3$ terms are absent because $M_0' = 0$ and $R_3' = 0$ as a result of TRS.

The approximate quantization condition (44) works remarkably well when compared to the numerics up to $f_{\max}(n)$, see Fig. 6 and 7. We can therefore assume that this equation is qualitatively correct beyond its derived range of validity and up to fluxes of order $f \sim 1/2$. In terms of magnetic oscillations (we anticipate on section 5), it gives the following Landau plot (oscillation index $n$ as a function of $f^{-1}$)

$$n = N_0(\epsilon)f^{-1} - \frac{1}{2} + \frac{\chi_0'(\epsilon)}{2f^{-1}}, \quad (54)$$

and clearly shows a non-linear behavior $\propto 1/f^{-1}$ on top of the familiar linear slope $f^{-1} +$ constant, see Fig. 8.

## 4.2 Two-dimensional semi-Dirac Hamiltonian

We now consider the 2D semi-Dirac Hamiltonian describing the merging point of two Dirac cones with opposite chirality [34, 35]. We use the Roth-Gao-Niu relation to obtain LLs from response functions. The semi-Dirac Hamiltonian [34] is

$$H = \frac{\Pi_x^2}{2m}\sigma_x + v_y\Pi_y\sigma_y. \quad (55)$$

We use units such that $\hbar = 1$, $v_y = 1$ and $m = 1$ (the energy unit is therefore $mv_y^2$, and that of length is $\hbar/(mv_y)$). The zero field spectrum is $\epsilon_\pm(\boldsymbol{k}) = \pm\sqrt{k_x^4/4 + k_y^2}$, the density of states is $\rho_0(\epsilon) = C\sqrt{\epsilon}$ (when $\epsilon \geq 0$) with $C = \frac{\Gamma(1/4)^2}{4\pi^{5/2}} \approx 0.187857$. The integrated DoS is

$$N_0(\epsilon) = \frac{2C}{3}\epsilon^{3/2}. \quad (56)$$

This is the number of states below energy $\epsilon > 0$ in the conduction band. Onsager's quantization condition gives the following semiclassical LLs [34]:

$$\epsilon_n^{SC} = \left(\frac{3}{2C}\right)^{2/3}\left[B(n+\frac{1}{2})\right]^{2/3}. \tag{57}$$

We now improve on these LLs by using the Roth-Gao-Niu relation. We first truncate eq. (7) at second order in the field

$$(n+\frac{1}{2})B \approx N_0(\epsilon) + BM_0'(\epsilon) + \frac{B^2}{2}\chi_0'(\epsilon), \tag{58}$$

and wish to solve for the LLs $\epsilon_n$ using our knowledge of the magnetic response functions. As there is a single valley and time-reversal symmetry, we know that $M_0 = 0$. In addition, we computed the susceptibility from linear response theory [36] and obtained (for $\epsilon > 0$)[5]

$$\chi_0(\epsilon) = -(2\pi)^2 \frac{D}{\sqrt{\epsilon}} \approx -\frac{0.84}{\sqrt{\epsilon}} \text{ with } D = \frac{\Gamma(3/4)^2}{4\pi^{5/2}}, \tag{59}$$

so that

$$\chi_0' = \frac{(2\pi)^2 D}{2\epsilon^{3/2}}. \tag{60}$$

Therefore we have to solve the following equation to obtain the LLs (at next order in the semiclassical/small field expansion):

$$\epsilon^3 - (n+\frac{1}{2})B\frac{3}{2C}\epsilon^{3/2} + \frac{3\pi^2 D}{2C}B^2 = 0. \tag{61}$$

This is a quadratic equation in $\epsilon^{3/2}$. We find that

$$\epsilon_n = \epsilon_n^{SC}g(n) \text{ with } g(n) = \left(\frac{1}{2} + \frac{1}{2}\sqrt{1 - \frac{1}{3\pi(n+\frac{1}{2})^2}}\right)^{2/3}. \tag{62}$$

This gives $g(0) \approx 0.918$ (instead of 0.808 [34]), $g(1) \approx 0.992$ (instead of 0.994 [34]) and $g(n \geq 2) \geq 0.997$ (instead of $\sim 1$ [34]). At fixed $(n+\frac{1}{2})B$, $\epsilon_n^{SC}$ is a constant and $g(n) \propto B^0 + B^2 + \mathcal{O}(B^4)$. The above equation is equivalent to writing

$$\epsilon_n = \left(\frac{3}{2C}\right)^{2/3}[B(n+\gamma_n)]^{2/3}, \tag{63}$$

with

$$\gamma_n = \frac{n+\frac{1}{2}}{2}\left(1 + \sqrt{1 - \frac{1}{3\pi(n+\frac{1}{2})^2}}\right) - n = \frac{1}{2} - \frac{1}{12\pi n} + \mathcal{O}(n^{-2}). \tag{64}$$

---

[5]There is another calculation of the orbital susceptibility for the semi-Dirac model in the literature [37]. These authors find $\chi_0(\epsilon) \approx -\frac{0.05}{\sqrt{\epsilon}}$ (instead of $\chi_0(\epsilon) \approx -\frac{0.84}{\sqrt{\epsilon}}$). There are reasons to doubt this result as it was obtained from an approximate single-band susceptibility formula derived from the "exact" Fukuyama formula. First, the approximate single-band susceptibility does not apply to a two-band model such as the semi-Dirac Hamiltonian. Second, this approximate single-band formula is known to be wrong even in the case of a single-band tight-binding model as it differs from the exact Landau-Peierls result, as discussed in [22].

## 4.3 Gapped graphene bilayer

From the knowledge of the magnetic response functions of the gapped graphene bilayer, we now seek to obtain approximate LLs. This section can be seen as the reversed path (LLs from response functions) compared to what we did in section 3.2, in which we obtained response functions from exact LLs. This case was already studied by Gao and Niu [1], who considered a single valley, resulting in a small mistake in the $M_0'$ term[6] so that we feel it is worth treating here.

At zero order, the LLs $\epsilon_{n,\lambda,\xi}^{(0)}$ are solution of :

$$(n+\frac{1}{2})B = N_0(\epsilon) \quad \rightarrow \epsilon_n^{(0)} = \lambda\sqrt{\Delta^2 + (n+\frac{1}{2})^2\omega_0^2}. \tag{65}$$

with $N_0 = \frac{m}{2\pi}\sqrt{\epsilon^2 - \Delta^2}$, where $\omega_0 = \frac{2\pi B}{m}$. The large $n$ behavior of $\epsilon_n^{(0)}$ is the same as that of the exact LLs $\epsilon_{n,\lambda,\xi}$ given in eq. (15). The major difference is that $\epsilon_n^{(0)}$ is valley independent and therefore for $\lambda\xi = -1$ the $n = 0, 1$ levels are not degenerate, moreover they appear field dependent. More quantitatively, to this order the deviation from exact LLs is

$$\delta^{(0)} = (\epsilon_{n,\lambda,\xi})^2 - (\epsilon_{n,\lambda,\xi}^{(0)})^2 = (2\lambda\xi n + \frac{3}{4} + \lambda\xi)\omega_0^2 = \mathcal{O}(B), \tag{66}$$

at fixed $nB$. This gives $\epsilon_{n,\lambda,\xi}^{(0)} - \epsilon_{n,\lambda,\xi} = -\xi\omega_0\sqrt{1 - \frac{\Delta^2}{\Delta^2+(\omega_0 n)^2}} + \mathcal{O}(B^2) = \mathcal{O}(B)$, which up to the overall sign agrees with [1]. At first order, the LLs $\epsilon_{n,\lambda,\xi}^{(1)}$ are solution of

$$(n+\frac{1}{2})B = N_0(\epsilon) + BM_0'(\epsilon) \quad \rightarrow \epsilon_{n,\lambda,\xi}^{(1)} = \lambda\sqrt{\Delta^2 + (n+\frac{1}{2} + \lambda\xi)^2\omega_0^2}, \tag{67}$$

with $M_0' = -\lambda\xi$ [1]. To this order, the LLs acquire a valley index dependency such that for $\lambda\xi = -1$ the $n = 0$ and $n = 1$ levels are now degenerate, however they still keep an unphysical field dependence. To this order the deviation from exact LLs is:

$$\delta^{(1)} = (\epsilon_{n,\lambda,\xi})^2 - (\epsilon_{n,\lambda,\xi}^{(1)})^2 = -\frac{1}{4}\omega_0^2 + ... = \mathcal{O}(B^2). \tag{68}$$

This gives $\epsilon_{n,\lambda,\xi}^{(1)} - \epsilon_{n,\lambda,\xi} = \lambda\frac{\omega_0^2}{8\sqrt{\Delta^2+\omega_0^2 n^2}} + \mathcal{O}(B^3) = \mathcal{O}(B^2)$, which disagrees with [1]. It is remarkable that the structure $(n+\frac{1}{2} + \lambda\xi)^2 = n^2 + n(2\lambda\xi + 1) + \mathcal{O}(n^0)$ present in eq. (67) agrees with the exact LLs (15) of structure $(n+1+\lambda\xi)(n+\lambda\xi) = n^2 + n(2\lambda\xi+1) + \mathcal{O}(n^0)$ in the semi-classical limit, i.e. when $n \rightarrow \infty$. In contrast, the LLs at zeroth order, see eq. (65), have a structure $(n+\frac{1}{2}) = n^2 + n + \mathcal{O}(n^0)$, which does not feature the valley index $\xi$ dependence.

At second order, the LLs $\epsilon_{n,\lambda,\xi}^{(2)}$ are solution of $(n+\frac{1}{2})B = N_0(\epsilon) + BM_0'(\epsilon) + \frac{B^2}{2}\chi_0'(\epsilon)$ giving

$$\epsilon_{n,\lambda,\xi}^{(2)} = \sqrt{\Delta^2 + \omega_0^2(n+\frac{1}{2} + \lambda\xi)^2\left(\frac{1}{2} + \frac{1}{2}\sqrt{1 - \frac{1}{2(n+\frac{1}{2}+\lambda\xi)^2}}\right)^2}, \tag{69}$$

with $\chi_0' = \frac{\pi}{2m}\frac{1}{\sqrt{\epsilon^2-\Delta^2}}$ [1]. To this order the deviation from exact LLs is:

$$\delta^{(2)} = (\epsilon_{n,\lambda,\xi})^2 - (\epsilon_{n,\lambda,\xi}^{(2)})^2 = \frac{\omega_0^2}{64n^2} + ... = \mathcal{O}(B^4). \tag{70}$$

---

[6]See footnote 2.

The field dependency of the $n = 0, 1$ levels has been greatly reduced but is still present in contrast to the exact LLs. This gives $\epsilon_{n,\lambda,\xi}^{(2)} - \epsilon_{n,\lambda,\xi} = -\frac{\omega_0^2}{128n^2\sqrt{\Delta^2+\omega_0^2 n^2}} + \mathcal{O}(B^5) = \mathcal{O}(B^4)$, which disagrees with [1].

Summarizing, this example shows that the deviation $\delta^{(p)}$ is expected to be of order $B^{p+1}$ when $B \to 0$ at fixed $nB$. The fact that $\delta^{(2)}$ turns out to be of order $B^4$ rather than $B^3$ is due to the vanishing of $R_3'$ (see section III.B). The main difference between the exact and the approximate LLs concerns the $n = 0$ and 1 levels. The exact $n = 0, 1$ levels are degenerate and field independent, whereas the approximate ones become degenerate at high enough order but remain field dependent at any order. The fact that $\epsilon_n^{(p)} - \epsilon_n$ is generally of order $B^{p+1}$ is also discussed in [1].

## 5 Magnetic oscillations in metals

### 5.1 Generalities

Magnetic oscillations occur in different physical quantities in a metal when the magnetic field or the electron density is varied. They are due to the quantization of closed cyclotron orbits into LLs and their traversal by the Fermi energy [6]. Most notably they occur in the longitudinal magnetoresistance (as found by Shubnikov and de Haas) [38], in the magnetization (de Haas-van Alphen oscillations) [39], in the density of states, which can be measured by scanning tunneling microscopy (see e.g. [40]) or in a quantum capacitance (see e.g. [41]), and in other quantities [7].

Magnetic oscillations are typically analyzed assuming either that the chemical potential is fixed or that the number of charge carriers is fixed [42], in which case the chemical potential oscillates with the magnetic field [43]. Experimental systems are closer to one or the other limit depending on the precise setup (isolated sample, contacted sample, substrate, etc). In the following, for simplicity, we assume that the chemical potential is fixed and analyze magnetic oscillations under this assumption, having mainly DoS oscillations in mind.

Magnetic oscillations can be decomposed in harmonics [7] (see also below). In the low field limit, when the cyclotron frequency is smaller than the temperature or the disorder broadening of LLs, the oscillations in the DoS are dominated by a fundamental harmonic. The latter has a sinusoidal shape modulated by an amplitude that decays exponentially with the temperature and the disorder broadening as described by the Lifshitz-Kosevich [8] and the Dingle reduction factors [7]. The usual way to analyze the oscillations is to index the maxima (or minima, see Appendix C) of the sinusoid by an integer $n$ and to plot this integer as a function of the inverse magnetic field $1/B$. This is known as a Landau plot or as a Landau fan diagram, see Fig. 8. In simple cases, the oscillations are periodic in $1/B$ and this plot is well fitted by a straight line:

$$n \approx \frac{F(\epsilon)}{B} - \gamma_0. \tag{71}$$

The slope gives the oscillation frequency $F(\epsilon)$ in teslas and the intercept at $1/B \to 0$ gives the zero-field phase-shift $\gamma_0$, which in simple cases is either a half-integer ("normal electrons") or an integer ("Dirac electrons"). In more complicated cases, it appears that the plot is not a straight line meaning that the oscillations are no-longer periodic in $1/B$ and that the value found for the zero-field phase-shift is neither an integer nor half an integer and is therefore hard to interpret [4].

The theoretical approach to a Landau plot starts either from the exact LLs, which are rarely known, or more often from a semiclassical quantization condition as pioneered by Onsager and Lifshitz [2]. For a detailed account of magnetic oscillations in two-dimensional electron

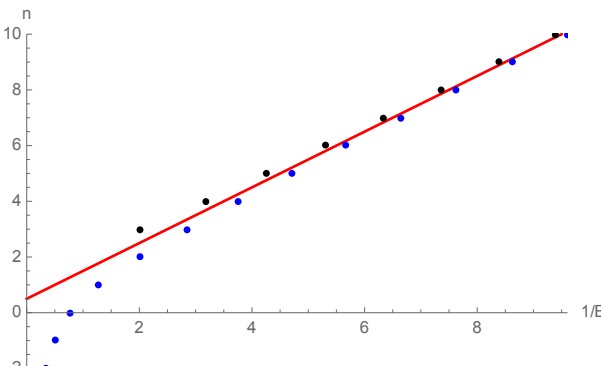

Figure 8: Sketch of a Landau plot or a Landau fan diagram analysis: the index $n$ used to label maxima in the magneto-conductance are plotted as a function of the inverse magnetic field $1/B$ (blue dots). A linear fit to the low field behavior is shown as a full red line. Deviation from linearity at high field are clearly visible. For this sketch, we used $n = 1/B + 1/2 - B$ for the blue dots, $n = 1/B + 1/2 + B$ for the black dots and $n = 1/B + 1/2$ for the red line [here $B$ is a dimensionless magnetic field]. The sign of the curvature term in the Landau plot $n = \text{function}(1/B)$ is given by that of $\chi'_0$. The slope gives $F = 1$ and the intercept at $1/B = 0$ gives $-\gamma_0 = 1/2$.

gases, see Champel and Mineev [44]. In particular, this reference discusses consequences on magnetic oscillations in thermodynamic quantities due to fixing the number of electrons rather than the chemical potential. For a general reference on magnetic oscillations, see the classic book by Shoenberg [7].

## 5.2 Aperiodic magnetic oscillations and non-linear Landau plot

Here, we will analyze the Landau plot starting from the Roth-Gao-Niu quantization condition, which appears to be more general than other semiclassical quantization conditions, and resolves certain issues especially concerning the phase of magnetic oscillations and the deviation from linearity of the Landau plot. For simplicity, we assume that there is a single species (valley, spin, etc.) here.

The Roth-Gao-Niu relation can be rewritten in an Onsager-like form

$$S(\epsilon) = (2\pi)^2 B[n + \gamma(\epsilon, B)], \tag{72}$$

where $S(\epsilon) = (2\pi)^2 N_0(\epsilon)$ is the reciprocal space area of a closed cyclotron orbit at energy $\epsilon$ and $\gamma$ is a phase-shift given by:

$$\gamma(\epsilon, B) = \frac{1}{2} - M'_0(\epsilon) - \frac{B}{2}\chi'_0(\epsilon) - \sum_{p \geq 3} \frac{B^{p-1}}{p!} R'_p(\epsilon). \tag{73}$$

This shows that the phase-shift $\gamma$ is in general a function of both the Fermi energy $\epsilon$ and the magnetic field $B$ (see also Wright and McKenzie [4]) and that it has an expression as a power series in the field $B$.

Using (72) instead of Onsager's relation $S(\epsilon) = (2\pi)^2 B[n + 1/2]$ in the standard derivation of the magnetic oscillations for the DoS (see in particular equations 2.161 and 2.163 in [7]) gives:

$$\rho_{\text{exact}}(\epsilon, B) = \rho_0(\epsilon) \left[ 1 + \sum_{p=1}^{\infty} 2\cos[2\pi p \left( \frac{N_0(\epsilon)}{B} - \gamma(\epsilon, B) \right)] \right]. \tag{74}$$

This means that the magnetic oscillations are no longer $1/B$-periodic because of the $B$ dependence of the phase-shift $\gamma(\epsilon, B)$. If $\gamma(\epsilon, B) = \gamma_0(\epsilon)$ does not depend on the magnetic field, then

the oscillations are strictly $1/B$-periodic with a period $1/N_0(\epsilon)$ but harmonics $p = 1, .., \infty$ are phase shifted by $2\pi p \gamma_0(\epsilon)$, see also [17]. It is also possible to integrate the above equation to obtain the IDoS including magnetic oscillations:

$$N_{\text{exact}}(\epsilon, B) = N_0(\epsilon) + \frac{B}{\pi} \sum_{p=1}^{\infty} \frac{1}{p} \left[ \sin[2\pi p \left( \frac{N_0(\epsilon)}{B} - \gamma(\epsilon, B) \right)] + \sin[2\pi p \gamma(\epsilon, B)] \right]. \quad (75)$$

For example, in the case of electrons in the continuum, we have $N_0(\epsilon) = \frac{m\epsilon}{2\pi B}$ and $\gamma(\epsilon, B) = \frac{1}{2}$ and we recover equation (93).

Inverting equation (72), we obtain [4]:

$$n = \frac{N_0(\epsilon)}{B} - \gamma(\epsilon, B) = \frac{N_0(\epsilon)}{B} - \gamma(\epsilon, 0) - B \frac{\partial \gamma}{\partial B}(\epsilon, 0) + \mathcal{O}(B^2). \quad (76)$$

The first term in equation (76) (the slope of $n$ as a function of $B^{-1}$) gives the frequency $F$ of magnetic oscillations as:

$$F(\epsilon) = N_0(\epsilon). \quad (77)$$

The frequency of oscillations is directly related to the density of carriers $n_c$ as $F(\epsilon) = \phi_0 n_c$ where $\phi_0 = h/e$ is the flux quantum (if there were several species, the frequency would be $F(\epsilon) = \phi_0 n_c / n_s$ where $n_s$ is the number of species). The second term gives the phase-shift in the zero field limit [1]:

$$\gamma_0(\epsilon) \equiv \gamma(\epsilon, 0) = \frac{1}{2} - M_0'(\epsilon). \quad (78)$$

The third term gives the deviation from linearity (or curvature) in the Landau plot:

$$C(\epsilon) = -\frac{\partial \gamma}{\partial B}(\epsilon, 0) = \frac{\chi_0'(\epsilon)}{2}. \quad (79)$$

In the end, the expansion reads

$$n = \frac{F(\epsilon)}{B} - \gamma_0(\epsilon) + C(\epsilon)B + ..., \quad (80)$$

which is a convenient expression to fit experiments on magnetic oscillations [4,45] (in ref. [4], $F$ is called $B_0$, $-\gamma_0$ is called $A_1$ and $C$ is called $A_2$). The low field expansion is valid if $B \ll \frac{F(\epsilon)}{\gamma_0(\epsilon)}$ (if $\gamma_0(\epsilon) \neq 0$). When $\gamma_0(\epsilon) \neq 0$, typically $\gamma_0(\epsilon)$ is of order 1 and the criterion is just $B \ll F(\epsilon)$. When $\gamma_0(\epsilon) = 0$, one needs to consider the next order to obtain the validity criterion, i.e. $n = \frac{F(\epsilon)}{B} + C(\epsilon)B + ...$ and $B \ll \sqrt{\frac{F(\epsilon)}{|C(\epsilon)|}}$.

The deviations from linearity are actually even richer in principle, see equation (9). The general structure being that of a series in power of $B$ in addition to the usual $1/B$ term: $n = \sum_{p=-1}^{\infty} c_p B^p$.

We also note that, apart from the frequency of oscillations that only depends on the zero-field energy spectrum, all other terms involve the geometry of Bloch states (or equivalently the energy spectrum in a finite field).

## 5.3 Phase of magnetic oscillations: Berry phase and average orbital magnetic moment

The second term of the above equation (76) for $n$ as a function of $\epsilon$ and $B$ can be easily related to a usual expression in terms of the Berry phase and of the orbital magnetic moment (see also [1]). We use the expression of the magnetization as a function of the Berry curvature

and of the orbital magnetic moment (this is the expression of the so-called "modern theory of magnetization", see [9] for review and also [46])

$$M_0(\epsilon) = \sum_{\alpha,\boldsymbol{k}} [\mathcal{M}_\alpha(\boldsymbol{k}) + \Omega_\alpha(\boldsymbol{k})(\epsilon - \epsilon_\alpha(\boldsymbol{k}))] \Theta(\epsilon - \epsilon_\alpha(\boldsymbol{k})), \tag{81}$$

where here $\alpha$ is a band index, $\boldsymbol{k}$ is a Bloch wavevector in the first Brillouin zone, $\mathcal{M}_\alpha(\boldsymbol{k})$ is the orbital magnetic moment and $\Omega_\alpha(\boldsymbol{k})$ is the Berry curvature. The Heaviside step function comes from the zero temperature limit of the Fermi function $n_F(\epsilon_\alpha) = \frac{1}{e^{(\epsilon_\alpha - \epsilon)/T} + 1}$. Differentiating with respect to the chemical potential

$$M_0'(\epsilon) = \sum_{\alpha,\boldsymbol{k}} [\mathcal{M}_\alpha(\boldsymbol{k})\delta(\epsilon - \epsilon_\alpha(\boldsymbol{k})) + 2\pi\Omega_\alpha(\boldsymbol{k})\Theta(\epsilon - \epsilon_\alpha(\boldsymbol{k}))] = \langle\mathcal{M}\rangle_\epsilon \rho_0(\epsilon) + \frac{\Gamma(\epsilon)}{2\pi}, \tag{82}$$

where $\rho_0(\epsilon) = \sum_{\alpha,\boldsymbol{k}} \delta(\epsilon - \epsilon_\alpha(\boldsymbol{k}))$ is the DoS,

$$\Gamma(\epsilon) = (2\pi)^2 \sum_{\alpha,\boldsymbol{k}} \Omega_\alpha(\boldsymbol{k})\Theta(\epsilon - \epsilon_\alpha(\boldsymbol{k})) = \sum_\alpha \int d^2k \,\Omega_\alpha(\boldsymbol{k})\Theta(\epsilon - \epsilon_\alpha(\boldsymbol{k})) \tag{83}$$

is the Berry phase [47] (for an iso-energy contour of energy $\epsilon$) and $\langle\mathcal{M}\rangle_\epsilon \equiv \rho_0(\epsilon)^{-1} \sum_{\alpha,\boldsymbol{k}} \mathcal{M}_\alpha(\boldsymbol{k})\delta(\epsilon - \epsilon_\alpha(\boldsymbol{k}))$ is the average of the orbital magnetic moment over the Fermi surface. Therefore:

$$\gamma_0(\epsilon) = \frac{1}{2} - M_0'(\epsilon) = \frac{1}{2} - \frac{\Gamma(\epsilon)}{2\pi} + \langle\mathcal{M}\rangle_\epsilon \rho_0(\epsilon). \tag{84}$$

Equation (84) is an important result, which was already obtained in several works, see Refs. [1,15,17,46,48,49]. The three terms in $\gamma_0(\epsilon)$ are: the Maslov index $1/2$ due to two caustics in the cyclotron orbit, the Berry phase $\Gamma(\epsilon)$ and the Wilkinson-Rammal (WR) phase $\langle\mathcal{M}\rangle_\epsilon \rho_0(\epsilon)$ due to the orbital magnetic moment. The last two terms occur at order $\hbar$ in the semiclassical quantization.

When inversion symmetry is present, one recognizes the often quoted result $\gamma_0 = \frac{1}{2} - \frac{\Gamma}{2\pi}$ between the phase of magnetic oscillations and the Berry phase [50]. It is then often assumed that $\gamma_0$ can only take two values, either $1/2$ for normal electrons (with zero Berry phase $\Gamma = 0$) or $0$ for massless Dirac electrons (with a quantized Berry phase of $\Gamma = \pi$). However, in general, the Berry phase $\Gamma(\epsilon)$ depends on the energy and is not quantized, see e.g. [15]. In addition, there is a second term in $\gamma_0$, the WR phase, involving the orbital magnetic moment. Note that these two terms (Berry phase and WR phase) both involve the cell-periodic Bloch wavefunctions and can not be obtained from the zero-field energy spectrum alone. The phase of magnetic oscillations therefore crucially depends on the geometry of the Bloch states.

We now give an alternative derivation of the above equation for $\gamma_0(\epsilon)$ following [15]. The quantization condition can be written as including only the Berry phase correction at the price of shifting the energy of the cyclotron orbit by a Zeeman-like term $-\mathcal{M}B$ due to the orbital magnetic moment:

$$S(\epsilon - \mathcal{M}B) = (2\pi)^2 B\left(n + \frac{1}{2} - \frac{\Gamma(\epsilon)}{2\pi}\right). \tag{85}$$

Expanding the l.h.s. at first order in $B$, we obtain

$$S(\epsilon - \mathcal{M}B) \approx S(\epsilon) - \langle\mathcal{M}\rangle_\epsilon BS'(\epsilon) = S(\epsilon) - \langle\mathcal{M}\rangle_\epsilon B4\pi^2\rho_0(\epsilon), \tag{86}$$

as $\rho_0(\epsilon) = N_0'(\epsilon)$ and $S(\epsilon) = 4\pi^2 N_0(\epsilon)$. Therefore

$$N_0(\epsilon) \approx B\left(n + \frac{1}{2} - \frac{\Gamma(\epsilon)}{2\pi} - \langle\mathcal{M}\rangle_\epsilon \rho_0(\epsilon)\right) = B[n + \gamma_0(\epsilon)], \tag{87}$$

which agrees with the above expression (84) for $\gamma_0(\epsilon)$.

In order to illustrate the above discussion of the phase of magnetic oscillations, we present two simple examples. First, in the case of doped graphene ($\epsilon > 0$), $\Gamma(\epsilon) = \pi$ does not depend on $\epsilon$, $\langle \mathcal{M} \rangle_\epsilon = 0$ and $\gamma(\epsilon, B) = 0$. Indeed, the Berry curvature is delta-like localized at $K$ and $K'$ points of the first Brillouin zone (it is like a very thin flux tube carrying a $\pi$ flux, i.e. half a flux quantum), and so is the orbital magnetic moment as $\mathcal{M}_\alpha(\mathbf{k}) = 2\pi\epsilon_\alpha(\mathbf{k})\Omega_\alpha(\mathbf{k})$ in a two-band model. Therefore, at finite doping $\epsilon > 0$, the average of the orbital magnetic moment over the Fermi surface indeed vanishes $\langle \mathcal{M} \rangle_\epsilon = 0$ but not the Berry phase $\Gamma(\epsilon) = \pi$ [15].

Second, in the case of boron nitride or gapped graphene, there is a staggered on-site energy $\pm\Delta$ that breaks the inversion symmetry of the honeycomb lattice so that the Berry curvature and the orbital magnetic moment are no longer delta-like. We consider a finite doping $\epsilon > \Delta$ with the Fermi level in the conduction band. One has an energy dependent Berry phase $\Gamma(\epsilon) = \pi(1 - \frac{\Delta}{\epsilon})$, an average orbital magnetic moment $\langle \mathcal{M} \rangle_\epsilon = -\frac{\pi}{\epsilon/v^2}\frac{\Delta}{\epsilon}$ which is non-vanishing (as the Berry curvature is $\Omega_+ = -\frac{\Delta v^2}{2\epsilon_+^3}$) and the density of states is $\rho_0(\epsilon) = \frac{|\epsilon|}{2\pi v^2}\Theta(|\epsilon| - \Delta)$ so that $\langle \mathcal{M} \rangle_\epsilon \rho(\epsilon) = -\frac{\Delta}{2\epsilon}$. Therefore the zero-field phase-shift $\gamma_0 = \frac{1}{2} - \frac{\Gamma(\epsilon)}{2\pi} + \langle \mathcal{M} \rangle_\epsilon \rho_0(\epsilon) = \frac{1}{2} - \frac{\pi}{2\pi}(1 - \frac{\Delta}{\epsilon}) - \frac{\Delta}{2\epsilon} = 0$ despite the fact that both the Berry phase and the orbital magnetic moment depend on $\epsilon$ [15].

In some simple cases, the zero-field phase-shift $\gamma_0$ is indeed quantized to either 0 or $1/2$ (modulo 1) independently of the energy. This is the case in the above examples of graphene, of boron nitride or of a graphene bilayer, for which it turns out that $\gamma_0 = \frac{1}{2} - \frac{W}{2}$ is independent of the energy and given by a winding number $W$ [15]. However, this is not generally the case even in the presence of particle-hole symmetry [4], as explained in [16].

### 5.4 From a measurement of magnetic oscillations to response functions

The measurement of magnetic oscillations as a function of the Fermi energy could in principle be used to obtain the (per species) magnetization and the susceptibility by integration [1]. One starts from a measurement of the position of maxima or minima in a physical quantity featuring magnetic oscillations to realize a Landau plot. This plot is then fitted by a relation such as

$$n = \text{function}(B, \epsilon) = \sum_{p=-1}^{\infty} c_p B^p . \tag{88}$$

and compared with equation (9). From the frequency $c_{-1}(\epsilon)$, one obtains the density of states by integration: $\rho_0(\epsilon) = \int^\epsilon d\epsilon c_{-1}(\epsilon) + \text{const}$. From the constant $c_0(\epsilon)$, one obtains the magnetization by integration: $M_0(\epsilon) = \int^\epsilon d\epsilon \left(\frac{1}{2} + c_0(\epsilon)\right) + \text{const}$. From the coefficient $c_1(\epsilon)$, one obtains the susceptibility by integration: $\chi_0(\epsilon) = 2\int^\epsilon d\epsilon c_1(\epsilon) + \text{const}$. Etc.

The main results in this section are first that the Landau plot is in general not a straight line but can be curved. This curvature is related to a response function $\chi_0'$ as given in equation (79). Second, the zero-field phase-shift extracted from a Landau plot not only depends on the Berry phase but also on the orbital magnetic moment (the so-called Wilkinson-Rammal phase) as given in equation (84).

## 6 Conclusion

The quantization condition recently proposed by Gao and Niu is a powerful generalization of the Onsager relation. It allows one to recover various known results scattered in the literature but also to establish new results. In this paper, we have explored several consequences of

this generalized Onsager condition, which we call Roth-Gao-Niu relation. In particular, we have shown how to analyze magnetic oscillations via Landau plots that are not necessarily linear and how to properly extract geometric quantities such as the Berry phase. Also, on several examples, we have shown how to improve analytically on semiclassical Landau levels. Or in the opposite direction, we have shown how to easily obtain information of magnetic response functions from exactly known Landau levels. We have also illustrated the limitation of the method on several examples in which singular behaviors such as Dirac delta functions or Heaviside step functions can not be captured.

## Acknowledgements

We would like to thank Mark Goerbig, Miguel Monteverde, Rémy Mosseri, Emilie Tisserond and Julien Vidal for useful discussions about magnetic oscillations. We also thank Hélène Bouchiat and Thierry Champel for insights on chemical potential oscillations and for indicating several references.

## A    Two-dimensional electron gas in the continuum

Here, we illustrate the decomposition of the exact grand potential (or DoS or IDoS) into a smooth part and a part containing the quantum magnetic oscillations in a case simple enough such that all calculations can be performed analytically: the two-dimensional electron gas in the continuum.

In general, the IDoS $N_{\text{exact}}(\epsilon, B)$ can be decomposed into a part at zero field (essentially a classical IDoS) $N_0(\epsilon)$, a part that depends on the field in a smooth manner $\delta N_{\text{smooth}}(\epsilon, B)$ and a part that depends on the field in an oscillating manner $N_{\text{osc}}(\epsilon, B)$. The smoothed IDoS $N(\epsilon, B)$ – which is called semiclassical IDoS by Gao and Niu [1] – is by definition the IDoS without the magnetic oscillations, therefore $N(\epsilon, B) = N_0(\epsilon) + \delta N_{\text{smooth}}(\epsilon, B)$. This is for example the result of a calculation using the Poisson summation formula or the Euler-MacLaurin formula (see [7], in particular appendix 3).

In the case of the two-dimensional electron gas in the continuum, the zero temperature grand potential defined by $\Omega_{\text{exact}}(\epsilon, B) = B \sum_n (\epsilon_n - \epsilon) \Theta(\epsilon - \epsilon_n)$ is [7]:

$$
\begin{aligned}
\Omega_{\text{exact}}(\epsilon, B) &= \Omega_0(\epsilon) + \frac{\pi B^2}{12m} + \frac{B^2}{\pi m} \sum_{p=1}^{\infty} \frac{1}{p^2} \cos\left[2\pi p \left(\frac{m\epsilon}{2\pi B} - \frac{1}{2}\right)\right] \\
&= \Omega_0(\epsilon) + \delta\Omega_{\text{smooth}}(\epsilon, B) + \Omega_{\text{osc}}(\epsilon, B).
\end{aligned} \tag{89}
$$

Gao and Niu's main step is the identification of the smoothed grand potential $\Omega(\epsilon, B) = \Omega_0(\epsilon) + \delta\Omega_{\text{smooth}}(\epsilon, B)$ with a certain limit of the grand potential at finite temperature. The latter is obtained from

$$
\Omega_{\text{exact}}(\mu, B, T) = -T N_\phi \sum_n \ln(1 + e^{-\beta(\epsilon_n - \mu)}) = \int d\epsilon [-n_F'(\epsilon)] \Omega_{\text{exact}}(\epsilon, B), \tag{90}
$$

where $n_F(\epsilon) = \frac{1}{e^{\beta(\epsilon - \mu)} + 1}$ is the Fermi-Dirac function with $\mu$ the chemical potential. Removing the magnetic oscillations is done by taking first the low field limit such that $T \gg \omega_c = |\epsilon_{n+1} - \epsilon_n|$ – where $\omega_c$ is the cyclotron frequency which is an increasing function of $B$ – and then the zero temperature limit $T \ll \mu$. In this way, the magnetic oscillations are washed out and the smoothed grand potential is $\Omega(\mu, B) = \lim_{T \ll \mu}\left[\lim_{T \gg \omega_c} \Omega_{\text{exact}}(\mu, B, T)\right]$.

Note that this is not the same as $\Omega_{\text{exact}}(\mu, B, T \to 0)$. The smoothed grand potential admits a series expansion in powers of $B$:

$$\Omega(\epsilon, B) = \Omega_0(\epsilon) - B M_0(\epsilon) - \frac{B^2}{2} \chi_0(\epsilon) - \sum_{p \geq 3} \frac{B^p}{p!} R_p(\epsilon). \tag{91}$$

As a consequence,

$$\delta\Omega_{\text{smooth}}(\epsilon, B) = \frac{\pi B^2}{12m} = -B M_0(\epsilon) - \frac{B^2}{2} \chi_0(\epsilon) - \sum_{p \geq 3} \frac{B^p}{p!} R_p(\epsilon), \tag{92}$$

and therefore we find that $R_p = 0$ for all $p \geq 1$ except $R_2 = \chi_0 = -\frac{\pi}{6m}$, which is the familiar Landau diamagnetism of a 2D electron gas.

From (89), the exact IDoS is obtained by $N = -\Omega'$ as

$$N_{\text{exact}}(\epsilon, B) = N_0(\epsilon) + \frac{B}{\pi} \sum_{p=1}^{\infty} \frac{1}{p} \sin\left[ 2\pi p \left( \frac{m\epsilon}{2\pi B} - \frac{1}{2} \right) \right] \tag{93}$$

so that the smoothed IDoS $N(\epsilon, B) = N_{\text{exact}}(\epsilon, B) - N_{\text{osc}}(\epsilon, B) = N_0(\epsilon)$ and $\delta N_{\text{smooth}}(\epsilon, B) = 0$ here.

# B  Historical remarks on the semiclassical quantization condition for cyclotron orbits

The semiclassical quantization condition reads $S(\epsilon) = (2\pi)^2 B[n + \gamma(\epsilon, B)]$ with

$$n + \gamma(\epsilon, B) = n + \frac{1}{2} - \frac{\Gamma(\epsilon)}{2\pi} - \langle \mathcal{M} \rangle_\epsilon \rho_0(\epsilon) - \frac{B}{2} \chi_0'(\epsilon) + \mathcal{O}(B^2). \tag{94}$$

The first two terms $(n + 1/2)$ were found by Onsager [2] from Bohr-Sommerfeld or WKB quantization of closed cyclotron orbits. The third term is the Berry phase [47]. That the Berry phase modifies the semiclassical quantization condition was understood by Kuratsuji and Iida [51] shortly after Berry's work. In the context of the quantization of closed cyclotron orbits, the Berry phase correction was found by Chang and Niu [52], Wilkinson and Kay [48], and also by Mikitik and Sharlai [50] as a consequence of the result of Roth [3]. The fourth term is the Wilkinson-Rammal phase related to the orbital magnetic moment. It was first found by Roth [3], then rediscovered by Wilkinson [54] and called Wilkinson-Rammal by Bellissard [30]. In an analysis of the Hofstadter butterfly, Wilkinson and Kay [48] and Chang and Niu [53] wrote a quantization condition which includes both the Berry phase and the Wilkinson-Rammal phase. The fifth term was also found by Roth [3], although in a less compact form.

Generally speaking, the semiclassical quantization of matrix-valued classical hamiltonians (for example a Bloch Hamiltonian $H(k_x, k_y)$) leads to the appearance of extra phases in the Bohr-Sommerfeld equation. Those phases are the Berry phase (related to the Berry curvature) and the Wilkinson-Rammal phase (related to the orbital magnetic moment). This is nicely discussed in Gat and Avron [46], that refer to the general work of Littlejohn and Flynn [55].

In the semiclassical approximation, the phase of the wavefunction is given by the reduced classical action $A_{\text{cl}}$ divided by $\hbar$, where $l_B = \sqrt{\frac{\hbar}{eB}}$ is the magnetic length [in this paragraph we restore units of $\hbar$ and $e$]. For a closed cyclotron orbit, $A_{\text{cl}} = S(\epsilon) l_B^2 \hbar$, where $l_B = \sqrt{\frac{\hbar}{eB}}$ is the magnetic length. Imposing that the wave-function be single-valued along a closed orbit leads to the quantization condition:

$$A_{\text{cl}} = \oint_{H=\epsilon} d\mathbf{r} \cdot \mathbf{p} = 2\pi\hbar \left( n + \frac{1}{2} - \frac{\Gamma(\epsilon)}{2\pi} - \frac{2\pi\hbar}{e} \langle \mathcal{M} \rangle_\epsilon \rho_0(\epsilon) - B \frac{\pi\hbar}{e} \chi_0'(\epsilon) + \mathcal{O}(B^2) \right). \tag{95}$$

The first term ($nh \sim A_{\mathrm{cl}}$) is classical, i.e. of order $\hbar^0$. The second (Maslov), third (Berry) and fourth (Wilkinson-Rammal) terms are of order $\hbar^1$ (e.g. Maslov is $\pi\hbar$). And the fifth term is of order $\hbar^2$. The Maslov index term occurs already for a scalar wave-function and is related to turning points (caustics). The other terms appear only for multi-component wave-functions.

## C  Maxima/minima in magneto-conductance/resistance, density of states and magnetization

The longitudinal magneto-conductivity generally has its maxima that coincide with that of the density of states, see e.g. [56, 57] and pages 153-155 in Shoenberg's book [7]. Counter-intuitively, this also corresponds to maxima in the longitudinal magneto-resistivity. This follows from the fact that the longitudinal resistivity is obtained by inverting the conductivity tensor and that the later has magnetic oscillations both in diagonal (longitudinal) and off-diagonal (Hall) elements [57].

Maxima in the DoS occur when the Fermi energy $\epsilon$ is exactly at the center of a LL $\epsilon_n$, i.e. when a LL is half-filled. Therefore $\epsilon = \epsilon_n$ or in other words when

$$N_0(\epsilon) = B[n + \gamma(\epsilon, B)]. \tag{96}$$

For the magnetization, the minima and maxima are shifted by a quarter period with respect to the density of states. See the discussion in Wright and McKenzie [4].

## D  Onsager quantization for electrons and for holes

The Onsager quantization condition for electrons reads

$$S(\epsilon) = (2\pi)^2 B(n + \frac{1}{2}), \tag{97}$$

where $S(\epsilon)$ is the $k$-space area of a closed cyclotron orbit of electron type. For holes, it is the $k$-space area of the hole orbit $S_h(\epsilon)$ which is quantized. Therefore it reads:

$$S_h(\epsilon) = S_{\mathrm{tot}} - S(\epsilon) = (2\pi)^2 B(n + \frac{1}{2}), \tag{98}$$

where $S_{\mathrm{tot}}$ is the $k$-space area when the band is full of electrons.

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
