# Peer review of "Landau levels, response functions and magnetic oscillations from a generalized Onsager relation"

_SciPost Physics, doi:SciPost Phys. 4, 024 (2018)_

## Round 2 · Referee Report · Anonymous · 2018-1-16

Strengths
1. Concrete suggestions for experimental testing of the new relation.
2. The paper has a good review character.
Weaknesses
Style
Report
The paper is a comprehensive study of applications of a generalized semiclassical-quantization condition, which was recently proposed by Gao and Niu. The authors consider various minimal models that describe the physics of condensed-matter systems of high current interest and compare the consequences of the new relation with results known from the literature. While the novelty lies in the original work by Gao and Niu, the main strength of this paper is that it provides rather concrete suggestions how to use the new relation, especially interesting for experimental studies. The main points for improvement I see in the style of the presentation. I think the paper should be published after the authors have addressed some suggestions listed below.
Requested changes
1. Below (40) I was missing a more detailed comparison with Ref. [26]. What are the possible reasons for the contradiction?
2. Typo in the in-text equation between (52) and (53): $k_x^2$ should be $k_x^4$.
3. Sentence below (67): "... an integer (normal electrons) or half an integer (Dirac electrons)." Shouldn't it be the other way around?
Minor stylistic changes:
4. Labeling subfigures.
5. Increasing the size of axes labels in some plots.
6. Between (5) and (6), the sentence "The prime denotes ..." should be placed closer to (5).
7. Review the manuscript with regard to grammer, especially punctuation.
(8., optional) The paper might profit from a table that summarizes the results.

---

## Round 2 · Referee Report · Anonymous · 2018-1-21

Strengths
1. it presents a systematic study
2. it presents various interesting case studies
3. its derivation is easy to follow
4. It is closely related to experiments
Weaknesses
1. some minor formatting issues
2. There are some points which can be explained better from my point of view
Report
This manuscript presents a systematic study of the connections between Landau levels and magnetic response functions based on a generalized Onsager's rule proposed in an earlier work. Using various interesting case studies, it explain in detail the three general applications of the generalized Onsager's rule. It also discusses the validity and weakness of the generalized Onsager's rule, to help accurately apply this rule. The last section about the magnetic oscillations in metals may be particularly interesting in experiments. In general, I find the manuscript interesting and recommend its publication.
I have a few questions/comments in the following.
(1) The formatting of Eq. 65 needs to be improved.
(2) In Page 13 below Eq.40, it is claimed that the susceptibility from the generalized quantization rule compares well with the result from Ref.29, but contradicts with Ref.26. I have read Ref.26, and found that Eq.4 in Ref.26 seems to have similar structure as Eq.39 in this manuscript. I hope the authors can explain more about this contradiction.
(3) In the second paragraph In Page 15, it is claimed that the fitting work well until the flux f=1/[2(n+1)]. Is this constraint f<=1/[2(n+1)] generally true?
Requested changes
No specific request.

---

## Round 2 · Referee Report · Anonymous · 2018-1-22

Strengths
1) The authors have assembled a wide range of case studies that present compelling evidence for the utility of a recent finding by Gao and Niu: that the corrections to the Bohr-Sommerfeld quantization rule may be identified as derivatives (with respect to energy) of zero-field response functions.
2) Given the importance of higher-order corrections to the quantization rule in interpreting high-field experiments, I believe that this work has merit and should be published in modified form.
Weaknesses
1) Ill-justified claims for novelty (which originate from an inaccurate interpretation of previous literature).
2) Misleading claims about the “limitations”/“difficulties” of the “Gao-Niu quantization rule”.
3) Oversimplified description of the method to extract the phase correction from quantum oscillations.
4) Failure to account for the field-dependence of the chemical potential in the analysis of the “nonlinear Landau plot”.
Report
The authors have assembled a wide range of case studies that present compelling evidence for the utility of a recent finding by Gao and Niu: that the corrections to the Bohr-Sommerfeld quantization rule may be identified as derivatives (with respect to energy) of zero-field response functions. Given the importance of higher-order corrections to the quantization rule in interpreting high-field experiments, I believe that this work has merit and should be published in modified form.
My suggestion for modifications are to (1) temper certain claims for novelty (which originate from an inaccurate interpretation of previous literature), (2-3) to reformulate two claims about misleadingly-labelled “limitations”/“difficulties” of the “Gao-Niu quantization rule”, (4) to clarify an oversimplified statement about the method to extract the phase correction from quantum oscillations, and finally (5) to account for the field-dependence of the chemical potential in the analysis of the “nonlinear Landau plot”.
Requested changes
1) While Gao and Niu deserve credit for formulating the single-band quantization rule in terms of response functions, I would point out that all corrections up to second order (including the magnetization and susceptibility terms) have been known since 1965 by L. R. Roth in Phys Rev 145 434. The authors recognize partially the importance of this earlier work, but cite her work erroneously in two locations:
1a) While authors recognize that Roth derived the orbital-magnetization correction to the quatization rule, they erroneously claim in footnote 52 that Roth did not derive the Berry phase term. In fact, the Berry term is present in Eq (41-42) of Phys Rev 145 434, which includes all corrections to second order. As the work was written in 1965, she did not, of course, call the correction a Berry phase. To see more explicitly the connection between Roth’s work and the Berry phase, I would point to arXiv:1710.04215. I would also point out that Mikitik – who was cited by present authors as identifying the Berry phase in cyclotron motion – actually credits Roth in his work.
1b) The authors suggest in their appendix ‘Historical Remarks’ that Roth only derived the Landau-Peierls orbital magnetic susceptibility (as a second order correction to the rule). This is incorrect; as stated earlier, Eq (41-42) contains the complete second-order correction. I believe the authors obtain this erroneous impression from Appendix 4 of Shoenberg’s monograph [which the authors cite], where Shoenberg offered an incomplete summary of Roth’s result “without interband effects”. Another reference that obtains the complete second-order correction – from a more traditional WKB perspective – is Fischbeck [Physica Status Solidi (b) 38, 11].
To recapitulate, Gao and Niu deserve credit for elegantly identifying already known corrections as derivatives of the magnetization and susceptibility.
As such, I am uncomfortable with how the authors’ claim that the “Gao-Niu quantization rule” is completely novel, disregarding the work of previous literature. For example, I feel the first sentence of abstract, and the second paragraph of the introduction, does not give Roth her deserved credit.
Minimally, I suggest to correct the above errors, and incorporate the at least part of the appendix of ‘Historical Remarks’ into the introduction, with much greater emphasis on Roth’s contributions.
As a tentative suggestion, the fairness of the name “Gao-Niu quantization condition” is debatable.
A related remark:
The authors claim Eq (79), which describes first-order corrections to the phase offset of quantum oscillations, is “one of their main results”. This is unacceptable, in the face of Roth’s pioneering work in of Phys Rev 145 434 [see, in particular Eq (41-42)]. In that work, Roth already discusses implications of the field-dependence of the phase offset in quantum oscillations.
2) My second remark regards a claim [point (iv) in page 5] that the authors have allegedly identified a “limitation” to the “Gao-Niu quantization condition”, in that it cannot be applied to energies where the quantities in the quantization condition become non-analytic in energy. The authors later identify these non-analyticities as originating from band edges in their case studies. I found overall the presentation of this fact to be misleading. It should not come as a surprise that semiclassical methods fail when the small parameter is no longer small; the small parameter here is 1/n, with n the Landau level index being of order one as one approaches a band edge. This failure is not specific to the Gao-Niu relation (as the authors seem to suggest), even the Onsager-Lifshitz condition without any corrections would fail near band edges. (In certain very simplified models, the semiclassical methods can be extended to small n. But this merely reflects the algebraic simplicity of certain models which admit exact solutions. In general, we should not expect semiclassical methods to be valid for small n; one should view large n as a consistency condition for the WKB approximation to be valid, at any order of approximation.)
3) Let me remark on another alleged “difficulty” of the Gao-Niu relation [point (v) in page 5]: that the Gao-Niu relation cannot be applied to “band contacts” such as Dirac points. There are a few issues here that are confusing. Are the authors claiming that the Gao-Niu relation cannot be applied at energies close to the Dirac point (cf. discussion in (ii) above) or are it cannot be applied to Dirac systems at all? Another complication is that the author proposes how to fix this “difficulty”: “add a gap, and let it vanish at the end of the calculation”. It is unclear if the authors now mean that the “difficulty” lies in the mathematical derivation of the Gao-Niu relation, rather than in the physical applicability. I suggest that the authors sharply delineate difficulties in mathematical derivation vs difficulties (or impossibilities) in physical application. I would guess that the authors are claiming a difficulty in mathematical derivation, but the current presentation misleads other readers into thinking the Gao-Niu relation cannot be applied to Dirac systems.
4) From page 19, it is claimed that: “The usual way to analyze the oscillations is to index their maxima (or minima) by an integer n and to plot this integer as a function of the inverse magnetic field 1/B.” While I agree that many experimental groups do this, I question the validity of this analysis. The Lifshitz-Kosevich formula for quantum oscillations is much more complicated than is suggested by the simplified statement above. In general, there is a sum over many harmonics. For large temperature (kT >> cyclotron energy) or large disorder ( lifetime *cyclotron frequency << 1), the dominant harmonic is the fundamental harmonic. Even so, the fundamental harmonic is not a simple trigonometric function, it has additional multiplicative factors which are field dependent (an exponential due to the finite lifetime, a sinh function due to finite temperature). To extract the phase offset with a Landau plot, what is really desired is extrema of the trigonometric function without the prefactors. In general, the Landau plot is meaningless for intermediate disorder and temperature. Here, multiple harmonics are important and the phase offset can only be extracted by fitting experimental data to the Lifshitz-Kosevich formula. The general way to extract these phase offsets, for closed orbits of any energy degeneracy and symmetry, is described in https://arxiv.org/abs/1707.08586.
5) Regarding Section 5.2 on the “nonlinear Landau plot”, the authors discuss how linear-in-B corrections to the phase shift (gamma) result in a “nonlinearity” in the Landau plot, with presumed application to the Schubnikov-de Haas and de Haas-van Alphen effects. While I do not contest the asymptotic character of gamma and its undeniable effect on quantum oscillations, I would point out that any Lifshitz-Kosevich formula for quantum oscillations is evaluated at a chemical potential that is field-dependent (due to fixed particle density, which is the experimental reality). The field-dependence of the chemical potential effectively introduces an additional field-dependence to the measured phase offset of quantum oscillations. To appreciate this, I offer the following argument. Consider that any Lifshitz-Kosevich formula receives as input the quantitity:
L^2 S(mu) + gamma(mu,B)
With L the magnetic length, S the area of an (extremal) orbit, and mu the chemical potential. Presuming mu is a function of B about an average (mu_ave), we might expand the above as
L^2 S(mu_ave) + L^2 dS/dmu (mu – mu_ave) + gamma
Mu-mu_ave is of order B in strictly 2D systems, and of order B^{3/2} in 3D systems [see Shoenberg, Magnetic oscillations in metals, equation 2.160]. We see that in 3D systems, the field-dependence of mu results effectively in a B^{1/2} dependence of phase offset, which is a stronger effect than the B-dependence that the authors consider. In strictly 2D systems, the field-dependence of mu results in a B^0 correction to phase offset, which essentially makes even the Berry phase (etc) unmeasurable. To recapitulate, I believe that the nonlinearities in Section 5.2 are missing a contribution which is actually larger in magnitude than the one considered by the authors.
Some minor technical points:
6) There are several locations where the authors claim there exists an analytic expansion in powers of the field (B). This would be highly surprising. WKB methods generically produce an asymptotic expansion in powers of the small parameter; for a discussion of this point (that is specific to the problem of a Bloch electron in a magnetic field), I recommend Blount [Phys Rev 126, 1636], or Roth [J. Phys. Chem. Solids 23, 433] or Fischbeck [Physica Status Solidi (b) 38, 11]. Also, I would point out that the Euler-Maclaurin approximation that underlies the Gao-Niu theory is generally not a convergent expansion (it is asymptotic).
In the case of cyclotron motion, I would offer the following physically-motivated argument that the expansion has to be asymptotic. My argument relies on the presence of magnetic breakdown (which is always present, though the significance of it may vary in different systems). Generally, the probability for breakdown goes like exponential[- #/B] [see, for example, pioneering work by Blount Phys Rev 126, 1636, and a more modern discussion in Phys. Rev. Lett. 119, 256601], which cannot be derived perturbatively. In other words, the assumption of analyticity implies that the quantization rule is valid for arbitrarily large B. However, we know for a fact that for large enough B, nonperturbative magnetic breakdown occurs. Even in the case of a single orbit, breakdown eventually occurs across different Brillouin zones (this fact was mentioned by the present authors, but its implications were not fully appreciated).
7) In page 3, the authors misleadingly refer to the second-order correction to the quantization condition as the derivative of an “orbital magnetic susceptibility”. The complete correction should also include a spin susceptibility (as the authors themselves point out in the case study of gapped graphene monolayer in page 6).
8) Regarding the “the winding number W of gapped graphene” in page 6, this should more appropriately be referred to as a winding number of a two-band approximation of gapped graphene (in fact, there are additional fine-tuned constraints to the two-band Hamiltonian, as the authors well know). I emphasize this point because the winding number loses its meaning for large enough Fermi energy (measured from the Dirac point).
9) Regarding “Whenever the energy spectrum in the presence of a magnetic field is the sum of an orbital part and of a spin part (i.e. without cross-terms) “, do the authors just mean that one component of spin is conserved? A more precise symmetry-based statement would be appreciated.
10) Regarding the model-specific claims that [page 6] “for systems with time-reversal symmetry…. upon summing [all odd derivatives] over both valleys, they vanish”, the authors may be interested in a model-independent proof of their claim (for the first derivative only) in Sec. IV-F of https://arxiv.org/abs/1707.08586.
11) Regarding the Rashba model of the surface states of a 3D topological insulator, the authors correctly claim that “it is usually assumed that only the inner circle matters”, however decline to explain why. I think it takes the mystery out of the description to offer a simple explanation.
12) In page 14, it looks like the authors are claiming that eta is the dimensionless parameter, but, as per the previous discussion, it should just be 1/n.
13) In page 15, “It is remarkable that the Gao-Niu corrections with respect to Onsager are important for all LLs and not just for those with small Landau index n ∼ 1”. I would actually expect that the semiclassical method should be valid for large n, not small.

---

## Round 3 · Author Response

Dear editor,

We appreciated the thorough reviewing of our paper by the three referees. We took all of their comments and remarks into account and modified our paper when appropriate. Below we reply to each of them. After this substantial revision, we now hope that our paper is suitable for publication Actually, in their first report, all three referees agreed on the fact that the paper should eventually be published.

Best regards,
Jean-Noël Fuchs, Frédéric Piéchon and Gilles Montambaux

Response to the first report:

We thank the referee for the careful reading and the positive report. The referee appreciated the review character and the discussion of concrete models that can be tested experimentally.

On the requested changes:

(1) We provided more details on the discrepancy between, on the one hand, our results and that of Suzuura and Ando PRB 2016, and, on the other hand, that of Schober et al., PRL 2012. We believe, the discrepancy is due to computational mistakes and not to any fundamental problem. The Fukuyama formula used by Schober et al. to compute the magnetic susceptibility should be valid when applied on such a separable Hamiltonian (no mixed k_x k_y term). We can not point to any precise mistake. But we know that their result for the susceptibility does not match the basic expectation for a Dirac cone (namely a diamagnetic delta-peak as found by McClure PR 1956), which is indeed obtained by Suzuura and Ando PRB 2016.

(2) Indeed there was a typo in the dispersion relation for the semi-Dirac model, that we corrected.

(3) The referee is right that this sentence should be the other way around. We corrected that.

(4) We labelled the subfigures systematically.

(5) We increased the size of labels in figures 6 and 7.

(6) We moved the sentence about the prime defining the derivative with respect to the Fermi energy closer to the relevant equation, as suggested by the referee.

(7) We systematically added punctuations in the equations, which was lacking. We also reviewed the complete paper to improve the grammar.

(8) We did not follow this suggestion of adding a table to summarize the results. We believe that the results are already summarized in the figures accompanying the different models.

Response to the second report :
We thank the referee for the positive report. The referee noted the clarity of presentation and the connection to experiments.

On the three points that are raised:

(1) The formatting of a few equations has been improved.

(2) We provided more details on the discrepancy between, on the one hand, our results and that of Suzuura and Ando PRB 2016, and, on the other hand, that of Schober et al., PRL 2012. We believe the discrepancy is due to computational mistakes and not to any fundamental problem. The Fukuyama formula used by Schober et al. to compute the magnetic susceptibility should be valid when applied on such a separable Hamiltonian. We can not point to any precise mistake. But we know that their result for the susceptibility does not match the basic expectation for a Dirac cone (namely a diamagnetic delta-peak as found by McClure PR 1956), which is indeed obtained by Suzuura and Ando PRB 2016.

(3) The restriction to fluxes f smaller than 1/[2(n+1)] is specific to the Hofstadter butterfly for the square lattice, but a similar argument could be applied to other lattices. It is related to the crossing between Landau levels emerging from the band bottom and their symmetric in energy from the band top. Indeed, in order to apply the semiclassical quantization, we require a broadened Landau level to carry a number of states which equals the Landau degeneracy (i.e. eB*A/h, where A is the sample area). From the Hofstadter butterfly, we see that the n th broadened LL encounters its energy symmetric partner when f=1/[2(n+1)]. Due to particle-hole symmetry, this encounter occurs at zero energy. For larger fluxes f, the broadened level starts depleting: the number of states it contains no longer equals the Landau degeneracy.
The precise value f=1/[2(n+1)] comes from previous studies of the Hofstadter butterfly which have shown that for a rational flux f=p/q with even denominator q=2k, the bands carrying number k and k+1 touch at zero energy (and with a linearly vanishing density of states, i.e. Dirac cones).
We modified the text in the article, to make the argument clearer.

Response to the third report:

We thank the referee for the thorough reviewing of our manuscript and for the many suggestions that helped us improve the paper. After this substantial revision, we now hope that the paper is suitable for publication.

On the requested changes:

(1) After reading Roth’s paper more carefully, we agree with most of the opinions of the referee.

(1a) Indeed the Berry phase term is potentially present in Roth’s formula for the phase shift gamma. As pointed out by the referee, this is clear from the fact that Mikitik and Sharlai 1999 have derived the Berry phase term from the Roth formula (first order term H_1). Let us however note that the relevant term in H_1 is gauge-dependent, can be non-periodic with the reciprocal lattice and is not present in every derivation of the first order effective Hamiltonian for a single band. We are more familiar with the work of Qian Niu and co-workers than with that of Roth 1962-1966, Kohn 1959 or Blount 1962. In their RMP 2010, Q. Niu and co-workers obtain an effective single band Hamiltonian that does not contain such a term at first order in the magnetic field. There, the Berry phase term comes from a modification of the commutator between the gauge-invariant position operator (the latter being defined as the canonical position operator plus the Berry connexion). Hence our (wrong) belief that Roth had missed the Berry phase correction to gamma. We adapted our paper in several places to account for that fact and to correctly credit Roth.

(1b) We also agree on the fact that the second order correction to gamma in Roth 1966 contains more than just the (derivative to the) Landau-Peierls susceptibility. It is claimed that it actually contains the total magnetic susceptibility including all inter-band corrections. This is hard to verify. We are well aware that there are many “exact formulas” of susceptibility for Bloch electrons (Hebborn-Sondheimer, Blount, Roth, Fukuyama, etc.). We also know that when testing these formulas on concrete multi-band examples, they sometimes give different answers and do not necessarily agree with a numerically exact procedure relying on the energy spectrum in a magnetic field (Hofsatdter butterfly). See Raoux, Piéchon, Fuchs and Montambaux, PRB 2015.
We removed the erroneous statements about Roth and updated the abstract and the introduction so as to emphasize her contributions. However, we still believe that there is a justification in using the name “Gao-Niu quantization condition”. First, it goes beyond second order in the magnetic field. Second, it is formulated in a quite different spirit than Roth’s contribution. It is more compact, more thermodynamic and actually also more useful (in the sense that it is easier to use).
Equation (79) in the previous version is indeed one of our main results even if we are by far not the first ones to obtain it (which we never claimed as should have been obvious from the fact that we cited several papers). We have changed this sentence to make it even clearer and have added other papers in which this relation was recently obtained (Mikitik and Sharlai PRB 2012; Alexandradinata et al. arXiv 2017; etc.).

(2) and (3): We agree with the referee that our discussion on the limitations of the Gao-Niu relation in what were previously points (iv) and (v) was confusing. We removed point (v) and rewrote point (iv). We do not say that the Gao-Niu relation does not apply to Dirac type systems (actually the examples we present often involve Dirac cones). What we mean is that certain type of singularities present in magnetic response functions (such as Dirac delta or Heaviside step functions) are not captured by the Gao-Niu relation. The fundamental issue here is in supposing that the grand potential Omega admits an expansion in non-negative integer powers of B. This fails, for example, when the chemical potential is tuned at some band edges or at some band contacts. There, the grand potential can behave as B^(3/2) (this is for example the case at epsilon=0 in graphene). Another situation in which this assumption fails is, as noted by the referee, in the case of magnetic breakdown.

(4) Shubnikov-de Haas oscillations are usually analyzed with a Landau plot. This was for example the case for graphene (see the original paper by Ph. Kim’s group in Nature 2005), the many examples cited in Wright and McKenzie PRB 2003 on surface states of topological insulators, Murakawa et al. Science 2013 on the giant Rashba system BiTeI, Tisserond et al. EPL 2017 on an organic salt featuring Dirac cones, etc. We are well aware that the Lifshitz-Kosevitch formula (and its extensions) includes more harmonics, the effect of disorder, the effect of temperature, etc. Here, we just wanted to point out that the phase and the frequency of the SdH oscillations are affected by the magnetic corrections to gamma.

(5) Actually, in transport experiments, the chemical potential is fixed (due to the contacts with the reservoirs that inject and collect electrons) rather than the number of electrons in the sample. We do not contest that for some thermodynamic quantities (for example dHvA oscillations in magnetization), measurements can be performed at fixed number of electrons in isolated samples, in which case the dependence of the chemical potential on the magnetic field becomes crucial to be taken into account (especially in 2D). This is well documented, for example in the paper by Champel and Mineev that we cite. Actually, in 2D, thermodynamic measurements are difficult and are the exception compared to transport measurements. In our paper, we now make it clear that we assume that the chemical potential is fixed and that we essentially have in mind the magnetic oscillations in the resistivity.
The Berry phase can be measured via SdH oscillations in 2D systems, as shown by the following two examples. First, the experiment of Ph. Kim (Nature 2005) on graphene shows the existence of 2D metals in which the chemical potential is supposed to be fixed. In this experiment, the phase of SdH oscillations is precisely obtained from a Landau plot. The Berry phase of pi for Dirac fermions in graphene was extracted in that way. As a second example, many SdH experiments have been performed on surface states of topological insulators (see Wright and McKenzie PRB 2013 for a good summary). In these experiments the non-linearities that we mention are clearly visible on Landau plots. In the end, we do not think that the dependance of the chemical potential on the magnetic field is an issue for transport experiments on 2D metals.

(6) We changed “analytic” to “power series”, which is actually what we meant. We do not argue about whether such a series is convergent or asymptotic.

(7) We agree with the referee and changed “orbital magnetic susceptibility” to “magnetic susceptibility”.

(8) We have added a sentence to clarify this point and cited relevant papers in which deviation from quantization of the gamma_0 appears when taking more general models to describe boron nitride (Wright and McKenzie PRB 2013, Goerbig et al., EPL 2014 and Alexandradinata et al. arXiv 2017).

(9) We just mean that the Hamiltonian is the sum of a part acting on the orbital degrees of freedom and a part acting on the spin degrees of freedom. We are not sure to understand what the referee is asking for.

(10) Time-reversal symmetry always implies that the magnetization vanishes M_0=0. Therefore, there should be no surprise in the fact that the first order response M’_0=0.

(11) We have clarified this point by specifying that in the case of surface states of topological insulators, the Rashba model is only a small wave-vector (rather than a low-energy) approximation. We have reformulated the corresponding sentence to “However, when describing surface states of topological insulators, the Rashba model is only valid for small wave-vectors and therefore only the inner circle should be considered”.

(12) We agree with the referee that the semiclassical parameter is roughly 1/n (which is actually more or less what was we had written, although not in a very transparent way). What we wanted to point out is that the expansion parameter is also f→0 such that (n+1/2)f = constant. We removed the confusing notation eta. In order to clarify the nature of the expansion and the small parameter, we added a discussion [now called point (v)] in section 2 on the validity of the Gao-Niu relation. We also added a short paragraph at the beginning of section 4 in order to discuss the general structure of LLs as obtained via the Gao-Niu relation (series in powers of B at fixed (n+1/2)B).

(13) We removed the sentence “It is remarkable that…. and not just for those with small Landau index n \sim 1” as we now feel that it was confusing.
Let us note, however, that both the Onsager and the Gao-Niu relations should be valid for large n or small f (at fixed f n). In the opposite limit (therefore small n or large f), Gao-Niu is expected to be closer to the exact result than Onsager. This is the reason for comparing them in this regime, where we expect them to disagree.
This can be seen on figure 6 upon considering a horizontal line at fixed energy E. At E=-2, for example, Gao-Niu LLs and Onsager LLs are indistinguishable for n=2 (small f), are close for n=1 (intermediate f), and are truly different for n=0 (large f).

---

## Round 3 · List of Changes

In addition to the requested changes by the referees, we have also made a few other changes, that we list here:
- we added several references to the bibliography concerning corrections to the Onsager semiclassical quantization: Chang and Niu PRL 1995 and PRB 1996; Mikitik and Sharlai PRB 2012; Alexandradinata et al. arXiv 2017
- in the section on the response function of bilayer graphene, we added a comparison to a (single valley) quadratic band contact point (and cite a relevant reference by Sun et al. PRL 2009). This example is interesting as it shows how two different but related systems (graphene bilayer versus checkerboard lattice) manage to respect time-reversal symmetry with either two or one valley, and with the same energy spectrum. Only the labelling of energy levels differs, but this matters when using the Gao-Niu relation.
- we modified the paragraph comparing the semi-classical LLs for the square lattice tight-binding model (Hofstadter butterfly) obtained either via the Onsager or the Gao-Niu relation.
- Figure 2c was changed as the density of states was plotted instead of M_0’

---

## Round 4 · Referee Report · Anonymous (Referee 3) · 2018-3-20

Report

The authors have addressed all but two of my concerns, which were labelled (4) and (5) in previous reports. I do not recommend publication until they are addressed.

4) The prevalent use of Landau plots does not guarantee their correctness. A Landau plot analysis (where only maxima or minima are plotted) presupposes that the fundamental harmonic is dominant. Such dominance occurs only for high temperature (kT >> cyclotron energy) and strong disorder ( cyclotron frequency*lifetime << 1). In particular, it is only the phase offset of the fundamental harmonic that is equal to the

\lambda=Berry phase + correction due to orbital moment + Maslov correction.

The phase offsets of higher harmonics are generally integer multiples of \lambda, as shown in the Lifshitz-Kosevich formulae in Phys. Rev. X 8, 011027. In general, the presence of multiple harmonics prevent any naïve extraction of the phase offset from identifying maxima (or minima):

a) It is possible that the reported “nonlinearities” in Landau plots are simply because of ignored higher harmonics in the interpretation of experimental data. The importance of higher harmonics has been emphasized by Dhillon and Shoenberg in http://rsta.royalsocietypublishing.org/content/248/937/1

b) Even if the fundamental harmonic were dominant, the “nonlinearities” may be a result of Dingle damping or finite temperature; both effects result in the fundamental harmonic not being a simple sinuisoidal function. The naïve method of extracting the phase offset presupposes (inaccurately) that the maxima of a sinuisoidal function is identified.

I recommend that the authors refine their Landau plot analysis and address (a-b) above. I am not against making simplifying assumptions, but in the current version of the manuscript the assumptions are not stated. One suggestion is to replace “In simple cases” with the assumptions of temperature and disorder stated above, and a direct statement that only the fundamental harmonic is assumed dominant. I would also like the authors to address point (b) above.

5) It is a misconception that a contact (0D or 1D) can change the chemical potential (an intensive property) of a 2D or 3D system. Attachment of a contact results merely in the creation of a contact potential difference, which compensates for the difference in the work functions of the two contacting conductors. The chemical potential in the bulk oscillates according to the thermodynamics of an electrically neutral system (i.e. a system with fixed density - not number - of electrons). These oscillations, in turn, may result in the oscillations of the contact potential, as described in Section 4.4 of Shoenberg’s “Magnetic oscillations in metals” and in the references of

https://www.sciencedirect.com/science/article/pii/0039602896004815

Further transport evidence of the field-dependence of the chemical potential in 2DEG can be found, e.g., in

http://journals.jps.jp/doi/10.1143/JPSJ.77.064713

I recommend that the authors retract their statement that the chemical potential of 2D metals is fixed, and address directly my concerns (stated in the previous report) that derive from the field-dependence of the chemical potential.

Minor comment: 1. In appendix C, it is claimed “generally” that the maximum of the longitudinal magneto-conductivity coincide with the maxima of the density of states. This claim is supported solely by reference 59 by Ando. However, I am a little puzzled by this citation. Ando’s work describes oscillations in the “transverse conductivity” (a different naming convention?). As far as I can see, Ando never directly claims that the maxima coincide, though he shows some suggestive numerical calculations, based on the self-consistent Born approximation and presumed short-range scatterers. All these restrictions suggest Ando’s claim is not “generally” valid. I recommend that the authors do more to support their “general” claim.

---

## Round 4 · Author Response

Dear editor,

We appreciated the thorough reviewing of our paper by the three referees. We took all of their comments and remarks into account and modified our paper when appropriate. Below we reply to each of them. After this substantial revision, we now hope that our paper is suitable for publication Actually, in their first report, all three referees agreed on the fact that the paper should eventually be published.

Best regards,
Jean-Noël Fuchs, Frédéric Piéchon and Gilles Montambaux

Response to the first report:

We thank the referee for the careful reading and the positive report. The referee appreciated the review character and the discussion of concrete models that can be tested experimentally.

On the requested changes:

(1) We provided more details on the discrepancy between, on the one hand, our results and that of Suzuura and Ando PRB 2016, and, on the other hand, that of Schober et al., PRL 2012. We believe, the discrepancy is due to computational mistakes and not to any fundamental problem. The Fukuyama formula used by Schober et al. to compute the magnetic susceptibility should be valid when applied on such a separable Hamiltonian (no mixed k_x k_y term). We can not point to any precise mistake. But we know that their result for the susceptibility does not match the basic expectation for a Dirac cone (namely a diamagnetic delta-peak as found by McClure PR 1956), which is indeed obtained by Suzuura and Ando PRB 2016.

(2) Indeed there was a typo in the dispersion relation for the semi-Dirac model, that we corrected.

(3) The referee is right that this sentence should be the other way around. We corrected that.

(4) We labelled the subfigures systematically.

(5) We increased the size of labels in figures 6 and 7.

(6) We moved the sentence about the prime defining the derivative with respect to the Fermi energy closer to the relevant equation, as suggested by the referee.

(7) We systematically added punctuations in the equations, which was lacking. We also reviewed the complete paper to improve the grammar.

(8) We did not follow this suggestion of adding a table to summarize the results. We believe that the results are already summarized in the figures accompanying the different models.

Response to the second report :
We thank the referee for the positive report. The referee noted the clarity of presentation and the connection to experiments.

On the three points that are raised:

(1) The formatting of a few equations has been improved.

(2) We provided more details on the discrepancy between, on the one hand, our results and that of Suzuura and Ando PRB 2016, and, on the other hand, that of Schober et al., PRL 2012. We believe the discrepancy is due to computational mistakes and not to any fundamental problem. The Fukuyama formula used by Schober et al. to compute the magnetic susceptibility should be valid when applied on such a separable Hamiltonian. We can not point to any precise mistake. But we know that their result for the susceptibility does not match the basic expectation for a Dirac cone (namely a diamagnetic delta-peak as found by McClure PR 1956), which is indeed obtained by Suzuura and Ando PRB 2016.

(3) The restriction to fluxes f smaller than 1/[2(n+1)] is specific to the Hofstadter butterfly for the square lattice, but a similar argument could be applied to other lattices. It is related to the crossing between Landau levels emerging from the band bottom and their symmetric in energy from the band top. Indeed, in order to apply the semiclassical quantization, we require a broadened Landau level to carry a number of states which equals the Landau degeneracy (i.e. eB*A/h, where A is the sample area). From the Hofstadter butterfly, we see that the n th broadened LL encounters its energy symmetric partner when f=1/[2(n+1)]. Due to particle-hole symmetry, this encounter occurs at zero energy. For larger fluxes f, the broadened level starts depleting: the number of states it contains no longer equals the Landau degeneracy.
The precise value f=1/[2(n+1)] comes from previous studies of the Hofstadter butterfly which have shown that for a rational flux f=p/q with even denominator q=2k, the bands carrying number k and k+1 touch at zero energy (and with a linearly vanishing density of states, i.e. Dirac cones).
We modified the text in the article, to make the argument clearer.

Response to the third report:

We thank the referee for the thorough reviewing of our manuscript and for the many suggestions that helped us improve the paper. After this substantial revision, we now hope that the paper is suitable for publication.

On the requested changes:

(1) After reading Roth’s paper more carefully, we agree with most of the opinions of the referee.

(1a) Indeed the Berry phase term is potentially present in Roth’s formula for the phase shift gamma. As pointed out by the referee, this is clear from the fact that Mikitik and Sharlai 1999 have derived the Berry phase term from the Roth formula (first order term H_1). Let us however note that the relevant term in H_1 is gauge-dependent, can be non-periodic with the reciprocal lattice and is not present in every derivation of the first order effective Hamiltonian for a single band. We are more familiar with the work of Qian Niu and co-workers than with that of Roth 1962-1966, Kohn 1959 or Blount 1962. In their RMP 2010, Q. Niu and co-workers obtain an effective single band Hamiltonian that does not contain such a term at first order in the magnetic field. There, the Berry phase term comes from a modification of the commutator between the gauge-invariant position operator (the latter being defined as the canonical position operator plus the Berry connexion). Hence our (wrong) belief that Roth had missed the Berry phase correction to gamma. We adapted our paper in several places to account for that fact and to correctly credit Roth.

(1b) We also agree on the fact that the second order correction to gamma in Roth 1966 contains more than just the (derivative to the) Landau-Peierls susceptibility. It is claimed that it actually contains the total magnetic susceptibility including all inter-band corrections. This is hard to verify. We are well aware that there are many “exact formulas” of susceptibility for Bloch electrons (Hebborn-Sondheimer, Blount, Roth, Fukuyama, etc.). We also know that when testing these formulas on concrete multi-band examples, they sometimes give different answers and do not necessarily agree with a numerically exact procedure relying on the energy spectrum in a magnetic field (Hofsatdter butterfly). See Raoux, Piéchon, Fuchs and Montambaux, PRB 2015.
We removed the erroneous statements about Roth and updated the abstract and the introduction so as to emphasize her contributions. The relation derived by Gao and Niu goes beyond second order in the magnetic field (which is what was obtained by Roth). Also, it is formulated in a quite different spirit than Roth’s contribution. It is more compact, more thermodynamic and actually also more useful (in the sense that it is easier to use). Nevertheless, we decided to call “Roth-Gao-Niu” the quantization condition at all orders in the field. We changed the paper accordingly.
Equation (79) in the previous version is indeed one of our main results even if we are by far not the first ones to obtain it (which we never claimed as should have been obvious from the fact that we cited several papers). We have changed this sentence to make it even clearer and have added other papers in which this relation was recently obtained (Mikitik and Sharlai PRB 2012; Alexandradinata et al. arXiv 2017; etc.).

(2) and (3): We agree with the referee that our discussion on the limitations of the Gao-Niu relation in what were previously points (iv) and (v) was confusing. We removed point (v) and rewrote point (iv). We do not say that the Gao-Niu relation does not apply to Dirac type systems (actually the examples we present often involve Dirac cones). What we mean is that certain type of singularities present in magnetic response functions (such as Dirac delta or Heaviside step functions) are not captured by the Gao-Niu relation. The fundamental issue here is in supposing that the grand potential Omega admits an expansion in non-negative integer powers of B. This fails, for example, when the chemical potential is tuned at some band edges or at some band contacts. There, the grand potential can behave as B^(3/2) (this is for example the case at epsilon=0 in graphene). Another situation in which this assumption fails is, as noted by the referee, in the case of magnetic breakdown.

(4) Shubnikov-de Haas oscillations are usually analyzed with a Landau plot. This was for example the case for graphene (see the original paper by Ph. Kim’s group in Nature 2005), the many examples cited in Wright and McKenzie PRB 2003 on surface states of topological insulators, Murakawa et al. Science 2013 on the giant Rashba system BiTeI, Tisserond et al. EPL 2017 on an organic salt featuring Dirac cones, etc. We are well aware that the Lifshitz-Kosevitch formula (and its extensions) includes more harmonics, the effect of disorder, the effect of temperature, etc. Here, we just wanted to point out that the phase and the frequency of the SdH oscillations are affected by the magnetic corrections to gamma.

(5) Actually, in transport experiments, the chemical potential is fixed (due to the contacts with the reservoirs that inject and collect electrons) rather than the number of electrons in the sample. We do not contest that for some thermodynamic quantities (for example dHvA oscillations in magnetization), measurements can be performed at fixed number of electrons in isolated samples, in which case the dependence of the chemical potential on the magnetic field becomes crucial to be taken into account (especially in 2D). This is well documented, for example in the paper by Champel and Mineev that we cite. Actually, in 2D, thermodynamic measurements are difficult and are the exception compared to transport measurements. In our paper, we now make it clear that we assume that the chemical potential is fixed and that we essentially have in mind the magnetic oscillations in the resistivity.
The Berry phase can be measured via SdH oscillations in 2D systems, as shown by the following two examples. First, the experiment of Ph. Kim (Nature 2005) on graphene shows the existence of 2D metals in which the chemical potential is supposed to be fixed. In this experiment, the phase of SdH oscillations is precisely obtained from a Landau plot. The Berry phase of pi for Dirac fermions in graphene was extracted in that way. As a second example, many SdH experiments have been performed on surface states of topological insulators (see Wright and McKenzie PRB 2013 for a good summary). In these experiments the non-linearities that we mention are clearly visible on Landau plots. In the end, we do not think that the dependance of the chemical potential on the magnetic field is an issue for transport experiments on 2D metals.

(6) We changed “analytic” to “power series”, which is actually what we meant. We do not argue about whether such a series is convergent or asymptotic.

(7) We agree with the referee and changed “orbital magnetic susceptibility” to “magnetic susceptibility”.

(8) We have added a sentence to clarify this point and cited relevant papers in which deviation from quantization of the gamma_0 appears when taking more general models to describe boron nitride (Wright and McKenzie PRB 2013, Goerbig et al., EPL 2014 and Alexandradinata et al. arXiv 2017).

(9) We just mean that the Hamiltonian is the sum of a part acting on the orbital degrees of freedom and a part acting on the spin degrees of freedom. We are not sure to understand what the referee is asking for.

(10) Time-reversal symmetry always implies that the magnetization vanishes M_0=0. Therefore, there should be no surprise in the fact that the first order response M’_0=0.

(11) We have clarified this point by specifying that in the case of surface states of topological insulators, the Rashba model is only a small wave-vector (rather than a low-energy) approximation. We have reformulated the corresponding sentence to “However, when describing surface states of topological insulators, the Rashba model is only valid for small wave-vectors and therefore only the inner circle should be considered”.

(12) We agree with the referee that the semiclassical parameter is roughly 1/n (which is actually more or less what was we had written, although not in a very transparent way). What we wanted to point out is that the expansion parameter is also f→0 such that (n+1/2)f = constant. We removed the confusing notation eta. In order to clarify the nature of the expansion and the small parameter, we added a discussion [now called point (v)] in section 2 on the validity of the Gao-Niu relation. We also added a short paragraph at the beginning of section 4 in order to discuss the general structure of LLs as obtained via the Gao-Niu relation (series in powers of B at fixed (n+1/2)B).

(13) We removed the sentence “It is remarkable that…. and not just for those with small Landau index n \sim 1” as we now feel that it was confusing.
Let us note, however, that both the Onsager and the Gao-Niu relations should be valid for large n or small f (at fixed f n). In the opposite limit (therefore small n or large f), Gao-Niu is expected to be closer to the exact result than Onsager. This is the reason for comparing them in this regime, where we expect them to disagree.
This can be seen on figure 6 upon considering a horizontal line at fixed energy E. At E=-2, for example, Gao-Niu LLs and Onsager LLs are indistinguishable for n=2 (small f), are close for n=1 (intermediate f), and are truly different for n=0 (large f).

---

## Round 4 · List of Changes

In addition to the requested changes by the referees, we have also made a few other changes, that we list here: - we added several references to the bibliography concerning corrections to the Onsager semiclassical quantization: Chang and Niu PRL 1995 and PRB 1996; Mikitik and Sharlai PRB 2012; Alexandradinata et al. arXiv 2017 - in the section on the response function of bilayer graphene, we added a comparison to a (single valley) quadratic band contact point (and cite a relevant reference by Sun et al. PRL 2009). This example is interesting as it shows how two different but related systems (graphene bilayer versus checkerboard lattice) manage to respect time-reversal symmetry with either two or one valley, and with the same energy spectrum. Only the labelling of energy levels differs, but this matters when using the Gao-Niu relation. - we modified the paragraph comparing the semi-classical LLs for the square lattice tight-binding model (Hofstadter butterfly) obtained either via the Onsager or the Gao-Niu relation. - Figure 2c was changed as the density of states was plotted instead of M_0’

---

## Round 5 · Author Response

Dear editor,

Below we reply to the referee’s second report. All his/her remaining concerns are related to magnetic oscillations, which constitute the third part (section 5) of our article. We think that we have now correctly answered all his/her criticisms contained in the first and second reports.

Best regards, The authors

Referee’s second report The authors have addressed all but two of my concerns, which were labelled (4) and (5) in previous reports. I do not recommend publication until they are addressed.

4) The prevalent use of Landau plots does not guarantee their correctness. A Landau plot analysis (where only maxima or minima are plotted) presupposes that the fundamental harmonic is dominant. Such dominance occurs only for high temperature (kT >> cyclotron energy) and strong disorder (cyclotron frequency*lifetime << 1). In particular, it is only the phase offset of the fundamental harmonic that is equal to the

\lambda=Berry phase + correction due to orbital moment + Maslov correction.

The phase offsets of higher harmonics are generally integer multiples of \lambda, as shown in the Lifshitz-Kosevich formulae in Phys. Rev. X 8, 011027. In general, the presence of multiple harmonics prevent any naïve extraction of the phase offset from identifying maxima (or minima):

a) It is possible that the reported “nonlinearities” in Landau plots are simply because of ignored higher harmonics in the interpretation of experimental data. The importance of higher harmonics has been emphasized by Dhillon and Shoenberg in http://rsta.royalsocietypublishing.org/content/248/937/1

b) Even if the fundamental harmonic were dominant, the “nonlinearities” may be a result of Dingle damping or finite temperature; both effects result in the fundamental harmonic not being a simple sinusoidal function. The naïve method of extracting the phase offset presupposes (inaccurately) that the maxima of a sinuisoidal function is identified.

I recommend that the authors refine their Landau plot analysis and address (a-b) above. I am not against making simplifying assumptions, but in the current version of the manuscript the assumptions are not stated. One suggestion is to replace “In simple cases” with the assumptions of temperature and disorder stated above, and a direct statement that only the fundamental harmonic is assumed dominant. I would also like the authors to address point (b) above.

Our answer to (4): The main goal of our article is not to recall how a Landau plot should be obtained experimentally (which is well known). However, we hear the criticisms of the referee and reply below.

That the phase offset in the case of a higher harmonic is a multiple of gamma_0(epsilon) = Berry + orbital moment correction + Maslov [called lambda by the referee] appears in our paper as equation (74). We acknowledge the fact that this is also present in Phys. Rev. X 8, 011027 (2018) by adding a citation to this reference.

(a) Magnetic oscillations usually contain several harmonics. In our paper, we make the simplifying assumption that the oscillations are dominated by the fundamental one. This is a reasonable approximation in the low field limit, when the cyclotron frequency is small compared to the temperature or to the disorder broadening. As suggested by the referee, we now clearly state these assumptions in the paper.

(b) Even in the case of a single harmonic, the Lifshitz-Kosevich formula shows that the magnetic field dependance is not only in a sinusoid but also in reduction factors R_T and R_D that account for thermal and disorder damping of the oscillations. This is well known and the reduction factors are routinely used by experimentalists to extract the cyclotron mass and the elastic scattering time (in the case of graphene, see for example Y. Zhang et al., Nature 2005 https://www.nature.com/articles/nature04235 and M. Monteverde et al., PRL 2010 https://journals.aps.org/prl/abstract/10.1103/PhysRevLett.104.126801). The main goal of our paper is not to recall the way a Landau plot should be obtained. However, as it seems to be an issue, we added a sentence to clarify that one should take the extrema of the oscillations after the extra magnetic-field dependence contained in the reduction factors such as R_T and R_D has been removed (i.e. one should take the extrema of the sinusoid only). Indeed, we are interested in sources of non-linearity in the Landau plot which are not related to thermal or disorder effects.

Referee: 5) It is a misconception that a contact (0D or 1D) can change the chemical potential (an intensive property) of a 2D or 3D system. Attachment of a contact results merely in the creation of a contact potential difference, which compensates for the difference in the work functions of the two contacting conductors. The chemical potential in the bulk oscillates according to the thermodynamics of an electrically neutral system (i.e. a system with fixed density - not number - of electrons). These oscillations, in turn, may result in the oscillations of the contact potential, as described in Section 4.4 of Shoenberg’s “Magnetic oscillations in metals” and in the references of

https://www.sciencedirect.com/science/article/pii/0039602896004815

Further transport evidence of the field-dependence of the chemical potential in 2DEG can be found, e.g., in

http://journals.jps.jp/doi/10.1143/JPSJ.77.064713

I recommend that the authors retract their statement that the chemical potential of 2D metals is fixed, and address directly my concerns (stated in the previous report) that derive from the field-dependence of the chemical potential.

Our answer to (5): After carefully re-reading the literature on magnetic oscillations in two-dimensional systems and interviewing colleagues (both experimentalists and theoreticians), we came to the conclusion that in 2D, nothing is simple and matter depend a lot on the precise experimental setup. Some systems are better described by a fixed density, some are better described by a fixed chemical potential, some are not described by these limits.

To contrast with the references given by the referee in his/her second report, we give below several references to the literature which support the opinion that in transport experiments with good contacts to the drain and source, the reasonable approximation is to assume that the chemical potential is fixed, not that the number of electrons or the density is fixed:

  • we cite from the discussion section in Sharapov, Gusynin and Beck, PRB 2004 https://journals.aps.org/prb/abstract/10.1103/PhysRevB.69.075104 about magnetic oscillations in graphene : “While for dHvA effect the condition [electron density] = const. is more natural, it is plausible that SdH effect can be measured under condition [chemical potential] = const.”

  • in the book by Shoenberg, there are also discussions about this issue in the case of 2D systems. On page 49 (in section 2.3.4 “Application to real 2D systems”): “It is not always clear whether the system more closely resembles one in which [the chemical potential] is constant or one in which [the density] is constant, as [the magnetic field] is varied.” Also page 157 on the quantum Hall effect in silicon MOSFET and in AsGa heterostructure: “These features can be most simply understood by supposing that for these samples it is [the chemical potential] rather than [the density] which remains constant as [the magnetic field] is varied, though this is probably a considerable oversimplification, and probably the interpretation for the MOSFET and for the heterostructure are somewhat different in detail, ...”

  • actually most experiments on SdH oscillations in 2D metals make the implicit assumption that the chemical potential is fixed, not the density. See again the two references on graphene that we cited above and also in our previous response to the first report: Y. Zhang et al., Nature 2005 https://www.nature.com/articles/nature04235 and M. Monteverde et al., PRL 2010 https://journals.aps.org/prl/abstract/10.1103/PhysRevLett.104.126801

In the end, we think that both assumptions (either constant chemical potential or constant density of electrons) are idealizations. Experiments are closer to one or the other limit depending on details of the setup and could actually be far from both limits. For example, in a transport experiment, contacts (to the drain and source reservoirs) can be ohmic (good contacts, chemical potential imposed by reservoir) or tunnel (bad contacts, almost isolated system, constant density of electrons). Both idealizations are equally questionable and there is no universal answer that would adapt to all experimental situations. In our work, for simplicity, we decided to stick to the grand canonical ensemble with fixed chemical potential. The fact that in some cases the chemical potential also oscillates with the magnetic field and renders the analysis of measurements more complicated need not concern us. We removed the statement about the chemical potential being fixed when the 2D system is contacted and rewrote the corresponding paragraph to clarify these issues and cite relevant references.

Referee: Minor comment: 1. In appendix C, it is claimed “generally” that the maximum of the longitudinal magneto-conductivity coincide with the maxima of the density of states. This claim is supported solely by reference 59 by Ando. However, I am a little puzzled by this citation. Ando’s work describes oscillations in the “transverse conductivity” (a different naming convention?). As far as I can see, Ando never directly claims that the maxima coincide, though he shows some suggestive numerical calculations, based on the self-consistent Born approximation and presumed short-range scatterers. All these restrictions suggest Ando’s claim is not “generally” valid. I recommend that the authors do more to support their “general” claim.

Our answer to the minor comment: Our claim is supported not only by ref 59 (Ando, see the post-scriptum below) but also by ref 60 (Coleridge et al., see equations 11 and 13). It is actually also supported by the reference (Endo and Iye, J. Phys. Soc. Jpn 2008) provided by the referee and from which we cite: “Our result suggest the relation [relative change in resistance proportional to relative change in density of states] remains valid regardless of the magnitude of [relative change in density of states]”.

A qualitative argument (attributed to Pippard 1965) is given in the book of Shoenberg in the SdH section 4.5 page 153: “He pointed out that the probability of scattering is proportional to the number of states into which the electrons can be scattered, and so this probability, which determines the electron relaxation time tau and the resistivity, will oscillate in sympathy with the oscillations of the density of states at the Fermi energy”. As an order of magnitude estimate, Shoenberg states on page 154 that the relative oscillations in the conductivity are given by the thermal reduction factor R_T multiplied by the relative oscillations in the DoS. This is essentially the content of equation (13) in ref 60 by Coleridge et al. There are many more similar statements in the book by Shoenberg. For example, in section 2.5 page 69: “… the oscillations of the density of states are closely related to the oscillations of resistivity (Shubnikov-de Haas effect)...”.

In the end, we do not have a mathematical proof but we believe that the statement is qualitatively and generally correct. We added two citations in the paper to give further support to our claim.

Concerning Ando’s paper (ref 59), transverse conductivity indeed means xx conductivity (usually called longitudinal) but it is transverse/perpendicular to the magnetic field hence the name "transverse conductivity" in this reference. The oscillations in the DoS (i.e. ImX=X’’, see equation (2.5)) are given in equation (2.14) and that in sigma_xx in equation (2.15). The comparison between the two equations shows that the maxima of the longitudinal magneto-conductivity coincide with the maxima of the density of states.

You are currently on this page

Resubmission 1712.02131v5 on 25 April 2018
Resubmission 1712.02131v3 on 22 February 2018

---

## Editorial Decision

published